# Melanocortin 1 receptor regulates cholesterol and bile acid metabolism in the liver

Keshav Thapa[1,2], James J Kadiri[1,2], Karla Saukkonen[1], Iida Pennanen[1], Bishwa Ghimire[3,4], Minying Cai[5], Eriika Savontaus[1,6,7], Petteri Rinne[1,6]*

[1]Research Centre for Integrative Physiology and Pharmacology, Institute of Biomedicine, University of Turku, Turku, Finland; [2]Drug Research Doctoral Programme (DRDP), University of Turku, Turku, Finland; [3]Institute for Molecular Medicine Finland (FIMM), HiLIFE Helsinki Institute of Life Science, University of Helsinki, Helsinki, Finland; [4]Faculty of Medicine, University of Turku, Turku, Finland; [5]Department of Chemistry and Biochemistry, University of Arizona, Tucson, United States; [6]Turku Center for Disease Modeling, University of Turku, Turku, Finland; [7]Unit of Clinical Pharmacology, Turku University Hospital, Turku, Finland

*For correspondence:
pperin@utu.fi

Competing interest: The authors declare that no competing interests exist.

**Abstract** Melanocortin 1 receptor (MC1-R) is widely expressed in melanocytes and leukocytes and is thus strongly implicated in the regulation of skin pigmentation and inflammation. MC1-R has also been found in the rat and human liver, but its functional role has remained elusive. We hypothesized that MC1-R is functionally active in the liver and involved in the regulation of cholesterol and bile acid metabolism. We generated hepatocyte-specific MC1-R knock-out (Mc1r LKO) mice and phenotyped the mouse model for lipid profiles, liver histology, and bile acid levels. Mc1r LKO mice had significantly increased liver weight, which was accompanied by elevated levels of total cholesterol and triglycerides in the liver as well as in the plasma. These mice demonstrated also enhanced liver fibrosis and a disturbance in bile acid metabolism as evidenced by markedly reduced bile acid levels in the plasma and feces. Mechanistically, using HepG2 cells as an in vitro model, we found that selective activation of MC1-R in HepG2 cells reduced cellular cholesterol content and enhanced uptake of low- and high-density lipoprotein particles via a cAMP-independent mechanism. In conclusion, the present results demonstrate that MC1-R signaling in hepatocytes regulates cholesterol and bile acid metabolism and its deficiency leads to hypercholesterolemia and enhanced lipid accumulation and fibrosis in the liver.

## Editor's evaluation

The significance of this manuscript is that it provides useful information for the field of hepatology and endocrinology on the regulatory mechanisms of cholesterol homeostasis by melanocortin. The authors provide solid evidence utilizing both in vivo and in vitro molecular, cellular, and biochemical approaches to support their claims.

## Introduction

Obesity is recognized as a global epidemic and is a major risk factor for type 2 diabetes, dyslipidemia, and cardiovascular disease (*Caballero, 2007*). In particular, visceral obesity is associated with atherogenic dyslipidemia characterized by high levels of triglycerides (TG), TG-rich lipoproteins, and low-density lipoprotein (LDL) cholesterol and reduced high-density lipoprotein (HDL) cholesterol in the

blood (*Grundy, 2004*; *Klop et al., 2013*). The liver plays a central role in this pathogenetic process as a regulator of cholesterol and fatty acid metabolism. As a consequence of hepatic dysregulation, de novo production and storage of cholesterol and fatty acids are enhanced leading to lipid accumulation in the liver, which eventually manifests as non-alcoholic fatty liver disease (NAFLD) (*Diehl and Day, 2017*; *Friedman et al., 2018*; *Than and Newsome, 2015*). Novel therapeutic strategies to enhance clearance and reduce excessive production of cholesterol and fatty acids in the liver are needed to mitigate the burden of obesity-associated dyslipidemia, NAFLD, and associated cardiovascular complications such as atherosclerosis.

Melanocortins are a family of peptide hormones that are proteolytically cleaved from the precursor molecule proopiomelanocortin (POMC) to yield adrenocorticotrophin (ACTH) and α-, β-, and γ-melanocyte stimulating hormone (α-, β-, and γ-MSH) (*Gantz and Fong, 2003*; *Smith and Funder, 1988*). Melanocortins bind to and activate five different G-protein coupled melanocortin receptor subtypes named from MC1-R to MC5-R (*Gantz and Fong, 2003*). Melanocortins and their receptors are expressed in the brain as well as in the periphery and regulate important physiological functions including skin pigmentation, sexual behavior, immune responses, and energy homeostasis (*Catania et al., 1996*; *Wikberg and Mutulis, 2008*; *Yeo et al., 2021*). There has been wide interest in melanocortin receptors as potential drug targets and in fact, melanocortin receptor-targeted drugs have been recently approved for the treatment of rare skin diseases and genetic obesity syndromes (*Montero-Melendez et al., 2022*). MC1-R was the first member of the melanocortin receptor family to be cloned and it binds only α-MSH with high affinity (*Mountjoy, 1994*; *Mountjoy et al., 1992*). MC1-R is abundantly expressed in melanocytes in the skin and is thus implicated as an integral regulator of skin pigmentation. MC1-R expression has been also demonstrated on a variety of peripheral cells including monocytes, macrophages, dendritic cells, neutrophils, endothelial cells, and fibroblasts (*Becher et al., 1999*; *Catania et al., 1996*; *Hartmeyer et al., 1997*; *Reichrath and Girndt, 2005*). Accordingly, increasing evidence demonstrates that MC1-R mediates potent and wide-ranging anti-inflammatory actions by suppressing the production of pro-inflammatory cytokines while simultaneously increasing the production of anti-inflammatory cytokines (*Catania et al., 2004*). Intriguingly, *MC1R/Mc1r* mRNA has also been detected in the human and rat liver following the testing of a hypothesis that MC-Rs might modulate inflammation in the liver (*Gatti et al., 2006*; *Malik et al., 2012*). However, these early studies aimed to only characterize the expression profile of different MC-Rs in the liver, and the functional role of hepatic MC1-R has thereby remained unexplored.

We have recently found that global deficiency of MC1-R signaling accelerates atherosclerosis in apolipoprotein E knockout mice (*Apoe$^{-/-}$*) by increasing arterial monocyte accumulation and by disturbing cholesterol and bile acid metabolism (*Rinne et al., 2018*). Specifically, MC1-R deficient mice showed elevated cholesterol levels in the plasma and liver in conjunction with a distinct bile acid profile characterized by reduced primary and increased secondary bile acid levels. This phenotype could be, however, caused by multiple mechanisms leaving an open question of whether MC1-R in the liver has a regulatory role in cholesterol and bile acid metabolism. In the present study, we aimed to address this question by engineering a hepatocyte-specific MC1-R knock-out mouse model. We here show that the loss of MC1-R signaling in hepatocytes causes hypercholesterolemia and enhanced lipid accumulation in the liver, and disturbs bile acid metabolism. Moreover, using HepG2 cells as an in vitro model, we found that α-MSH and selective MC1-R activation reduced cellular cholesterol content and enhanced uptake of LDL and HDL. This study demonstrates that MC1-R is functionally active in the liver and regulates cholesterol and bile acid metabolism in a protective way that could also have therapeutic implications.

## Results

### Hepatocyte-specific MC1-R deficiency enhances cholesterol and lipid accumulation in the liver

We first aimed to investigate whether MC1-R is expressed in the mouse liver. Immunohistochemical staining revealed a strong and uniform expression of MC1-R in the liver (*Figure 1A*). Immunofluorescence staining further revealed that MC1-R expression co-localizes with the hepatocyte marker serum albumin (*Figure 1B*) as well as with the cholangiocyte marker cytokeratin 19 and the monocyte and macrophage marker Mac-2 (*Figure 1—figure supplement 1*), while no clear colocalization

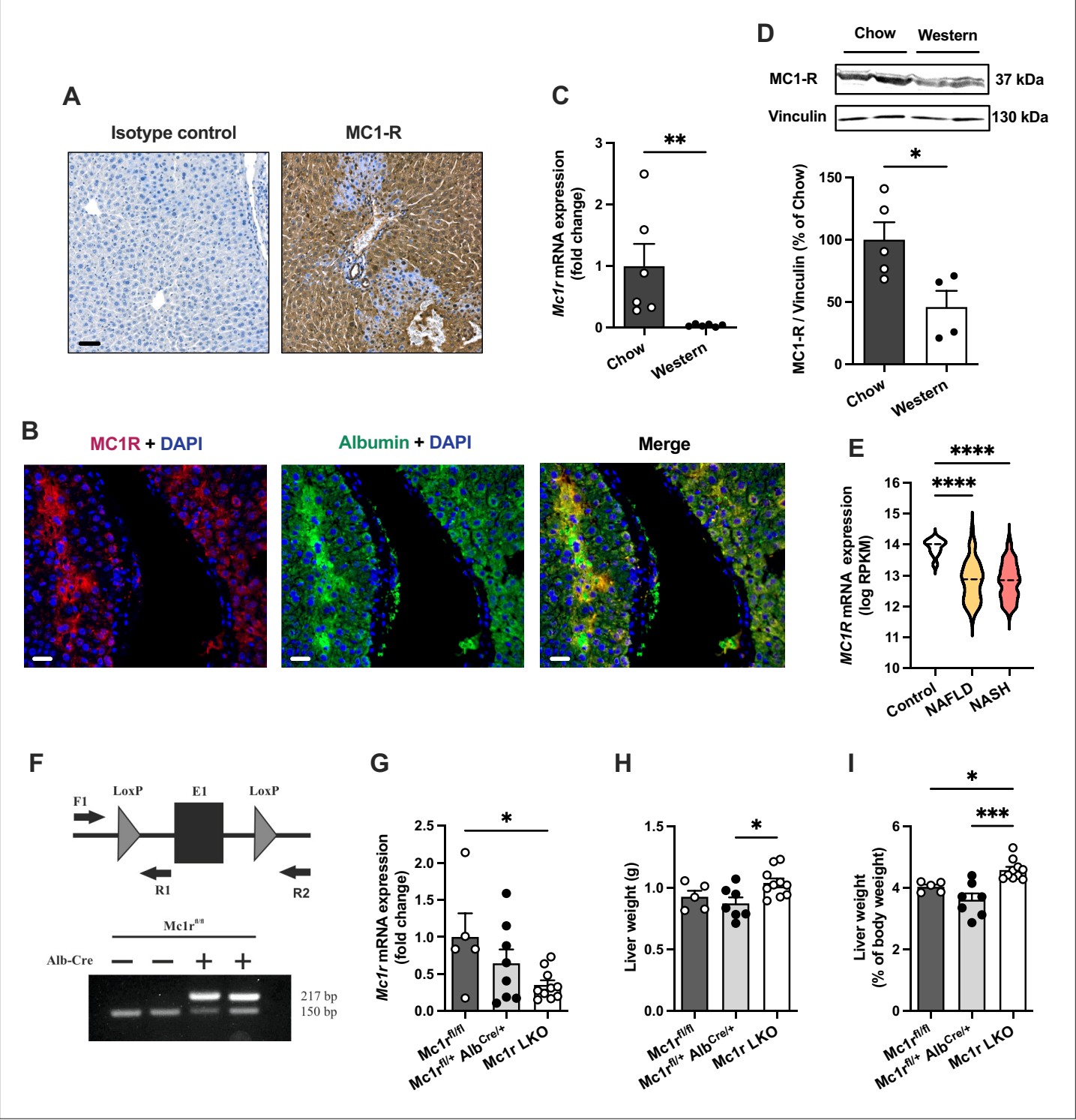

**Figure 1.** Melanocortin 1 receptor (MC1-R) is expressed in the mouse liver and down-regulated in mice fed a cholesterol-rich diet. (**A**) Immunostaining of MC1-R staining in the liver of chow-fed C57Bl/6 J mouse. In the control section, anti-MC1-R antibody was replaced by purified normal rabbit IgG (isotype control). Scale bar, 50 μm. (**B**) Immunofluorescence staining of MC1-R (red) and the hepatocyte marker serum albumin (green) in the liver of chow-fed C57Bl/6 J mouse. Scale bar, 20 μm. (**C**) Quantitative real-time polymerase chain reaction (qPCR) analysis of *Mc1r* mRNA expression in the liver of chow- and Western diet-fed mice. (**D**) Representative Western blots of MC1-R and β-actin (loading control) and quantification of MC1-R protein level in the liver of chow- and Western diet-fed mice. *p<0.05* and **p<0.01* versus chow-fed mice by Student's t-test. (**E**) *MC1R* gene expression in human liver biopsies from control cases (n=10) and patients with nonalcoholic fatty liver disease (NAFLD, n=51) or nonalcoholic steatohepatitis (NASH, n=155). Violin plots show normalized log2 RPKM values (reads per kilobase of exon per million reads mapped) and medians (dashed line) for each sample

*Figure 1 continued on next page*

*Figure 1 continued*

group. (**F**) Schematic presentation of the loxP-flanked (floxed) *Mc1r* allele and the positions of forward and reverse primers used for PCR genotyping. PCR analysis of genomic DNA extracted from the liver of Alb-Cre-negative and -positive mice that were homozygous for the Mc1r floxed allele (*Mc1r$^{fl/}$* $^{fl}$). The size of the recombined allele is ~217 bp. (**G**) qPCR analysis of *Mc1r* expression in the liver of chow-fed *Mc1r$^{fl/fl}$*, *Mc1r$^{fl/+}$ Alb$^{Cre/+}$*, and Mc1r LKO (*Mc1r$^{fl/fl}$ Alb$^{Cre/+}$*) mice at the age of 16 weeks. (**H, I**) Absolute liver weight and liver to body weight ratio (expressed as a percentage of body weight) in chow-fed *Mc1r$^{fl/fl}$*, *Mc1r$^{fl/+}$ Alb$^{Cre/+}$*, and Mc1r LKO mice at the age of 16 weeks. Values are mean ± SEM, n=5–10 mice per group in each graph. *p<0.05, **p<0.01, and ****p<0.0001 for the indicated comparisons by one-way ANOVA and Dunnet *post hoc* tests. Mc1r LKO, hepatocyte-specific MC1-R knock-out mice.

The online version of this article includes the following source data and figure supplement(s) for figure 1:

**Source data 1.** Uncropped Western blots for *Figure 1D*.

**Figure supplement 1.** Immunofluorescence staining of melanocortin 1 receptor (MC1-R) in the mouse liver.

**Figure supplement 2.** Pre-adsorption control for Melanocortin 1 receptor (MC1-R) Western blotting.

**Figure supplement 2—source data 1.** Uncropped Western blots for *Figure 1—figure supplement 2*.

**Figure supplement 3.** Hepatocyte-specific Melanocortin 1 receptor (MC1-R) deficiency does not affect body weight or composition in chow-fed female mice.

**Figure supplement 3—source data 1.** Uncropped Western blots for *Figure 1—figure supplement 3*.

was observed in CD31-positive endothelial cells (*Figure 1—figure supplement 1*). Furthermore, we sought to investigate whether the expression level of MC1-R in the liver is affected by feeding mice a cholesterol-rich Western diet. Remarkably, 12 weeks of Western diet feeding resulted in significant downregulation of the MC1-R mRNA level in the liver (*Figure 1C*). This result was further corroborated by Western blotting, which showed reduced protein expression of MC1-R in the liver of Western diet-fed mice (*Figure 1D* and *Figure 1—source data 1*). The specificity of the MC1-R signal was validated by pre-adsorption of the antibody with an MC1-R blocking peptide (*Figure 1—figure supplement 2* and *Figure 1—figure supplement 2—source data 1*). Furthermore, using RNA sequencing data from human liver biopsies (*Govaere et al., 2020*), we found that hepatic *MC1R* expression was significantly downregulated (log2 fold change = −1.1) in patients with NAFLD or nonalcoholic steatohepatitis (NASH) compared to control cases (*Figure 1E*).

To determine the regulatory role of MC1-R in the liver, we generated hepatocyte-specific MC1-R knock-out mice (*Mc1r$^{fl/fl}$ Alb$^{Cre/+}$*; denoted as Mc1r LKO) by crossing MC1-R floxed (*Mc1r$^{fl/fl}$*) mice with transgenic mice expressing Cre recombinase under the control of the mouse albumin promoter (*Alb$^{Cre/+}$*) (*Figure 1E*). Genotyping of the liver samples verified efficient recombination of the loxP-flanked allele in Mc1r LKO mice (*Figure 1E*) that resulted in significant downregulation of hepatic *Mc1r* mRNA expression in these mice compared to control *Mc1r$^{fl/fl}$* mice (*Figure 1F*). A gene dosage effect was also noted in this regard as *Alb$^{Cre/+}$* mice that were heterozygous for the *Mc1r* floxed allele (*Mc1r$^{fl/+}$ Alb$^{Cre/+}$*) showed only partial downregulation of *Mc1r* compared to Mc1r LKO mice (*Figure 1F*). Western blotting also showed significantly reduced MC1-R protein expression in the liver of Mc1r LKO mice (*Figure 1—figure supplement 3* and *Figure 1—figure supplement 3—source data 1*). To evaluate the effect of hepatocyte-specific MC1-R deficiency on body weight development, *Mc1r$^{fl/fl}$*, *Mc1r$^{fl/+}$ Alb$^{Cre/+}$*, and Mc1r LKO mice were fed a normal chow diet and weighed weekly during a monitoring period from 8 to 16 weeks of age. However, no differences were observed in body weight between the genotypes (*Figure 1—figure supplement 3*). Body composition analysis by quantitative NMR scanning at the start and end of the monitoring period did not reveal any significant changes in total fat or lean mass of Mc1r LKO mice compared to control genotypes (*Figure 1—figure supplement 3*). Of note, Mc1r LKO mice displayed a significant increase in liver weight (*Figure 1G*), which became more evident when calculated as the percentage of body weight (*Figure 1H*). Since the relative liver weight was significantly increased in comparison with both control groups, we used *Mc1r$^{fl/+}$ Alb$^{Cre/+}$* mice as the control group in subsequent analyses to eliminate the possible confounding by Alb-Cre transgene expression.

Histological examination by H&E and Oil Red O staining revealed an enhanced accumulation of intracellular lipid droplets in the liver of Mc1r LKO mice in comparison to the control (*Mc1r$^{fl/+}$ Alb$^{Cre/+}$*) mice (*Figure 2A*). Supporting this finding, quantification of hepatic lipid content showed increased TG and total cholesterol levels in Mc1r LKO mice (*Figure 2B and C*). Likewise, plasma TG and TC levels were significantly higher in Mc1r LKO mice compared to control mice (*Figure 2D and E*). Mc1r LKO

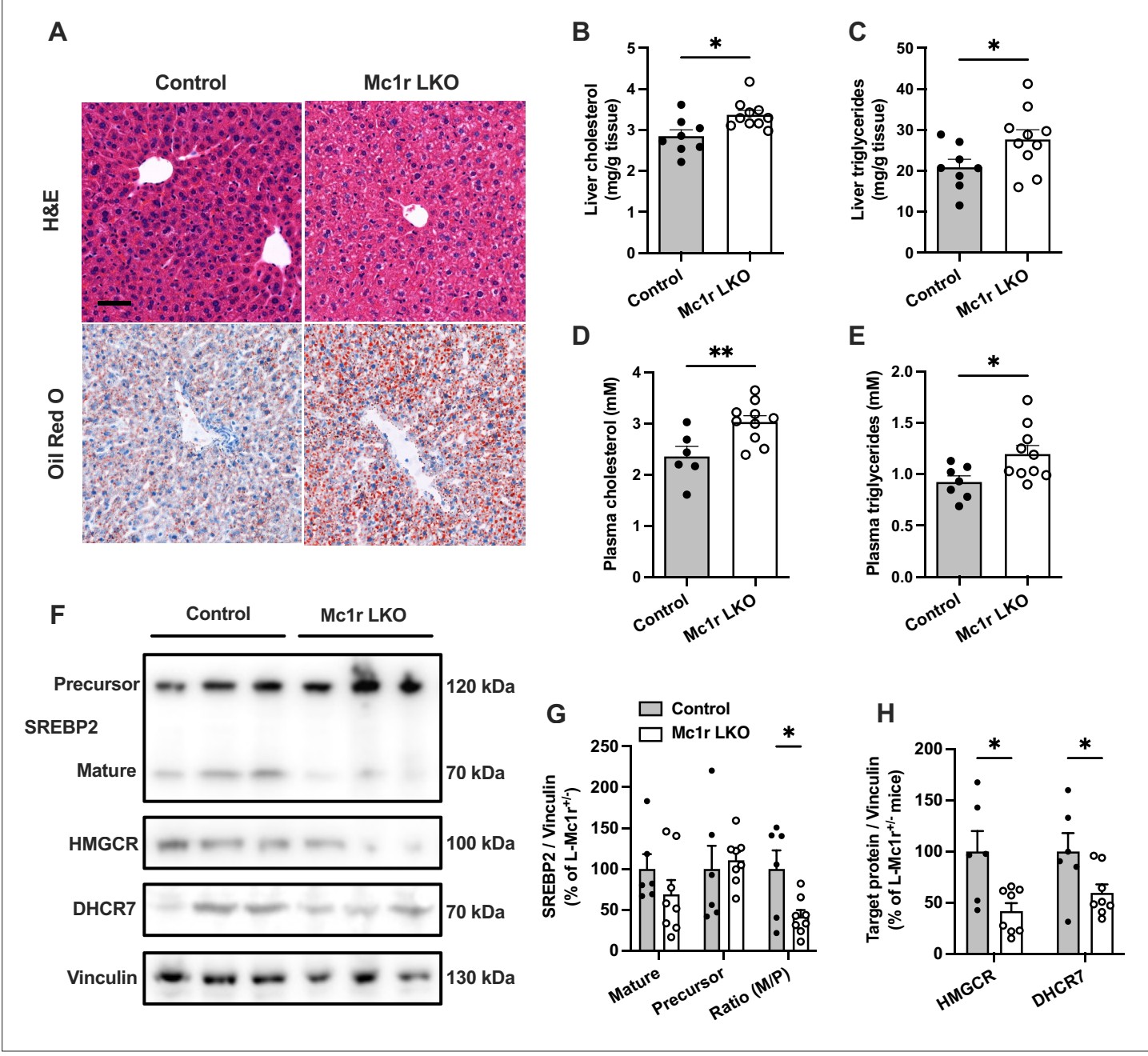

**Figure 2.** Hepatocyte-specific Melanocortin 1 receptor (MC1-R) deficiency enhances cholesterol and triglyceride accumulation in the liver. (**A**) Representative hematoxylin and eosin (H&E) and Oil Red O-stained liver sections of chow-fed control (*Mc1r^fl/+ Alb^Cre/+*) and Mc1r LKO mice. Scale bar, 50 µm. (**B, C**) Quantification of liver total cholesterol and triglyceride content in chow-fed control and Mc1r LKO mice. (**D, E**) Quantification of plasma total cholesterol and triglyceride concentrations in chow-fed control and Mc1r LKO mice. (**F**) Representative Western blots of SREBP2, HMGCR, DHCR7, and vinculin (loading control) expression in the liver of chow-fed control and Mc1r LKO mice. (**G**) Quantification of mature and precursor forms SREBP2 as well as their ratio (precursor-to-mature) in the liver of chow-fed control and Mc1r LKO mice. (**H**) Quantification of HMGCR and DHCR7 protein levels in the liver of chow-fed control and Mc1r LKO mice. Values are mean ± SEM, n=6–10 mice per group in each graph. *p<*0.05* and **p<*0.01* versus control mice by Student's t-test. SREBP2, sterol regulatory element binding protein 2; HMGCR, 3-hydroxy-3-methylglutaryl-CoA reductase; DHCR7, 7-dehydrocholesterol reductase.

The online version of this article includes the following source data and figure supplement(s) for figure 2:

**Source data 1.** Uncropped Western blots for *Figure 2F*.

**Figure supplement 1.** Hepatocyte-specific melanocortin 1 receptor (MC1-R) deficiency enhances liver fibrosis.

**Figure supplement 2.** Hepatocyte-specific melanocortin 1 receptor (MC1-R) deficiency does not affect body or liver weight in Western diet-fed female mice, but aggravates hepatic cholesterol and triglyceride accumulation.

mice also demonstrated signs of increased liver fibrosis, as evidenced by Picrosirius Red staining and gene expression analysis of fibrotic genes (*Figure 2—figure supplement 1*). However, no change in the expression of pro-inflammatory genes was observed between the genotypes (*Figure 2—figure supplement 1*). In a separate experiment, mice were fed a Western diet for 12 weeks to investigate whether hepatocyte-specific MC1-R deficiency exacerbates diet-induced dyslipidemia. Western diet-fed Mc1r LKO mice did not differ from control mice in terms of body weight or composition, liver weight, or plasma lipid concentrations (*Figure 2—figure supplement 2*). However, in line with the phenotype observed in chow-fed mice, Mc1r LKO mice showed enhanced cholesterol and TG accumulation in the liver after Western diet feeding (*Figure 2—figure supplement 2*).

We next quantified protein levels of genes involved in cholesterol synthesis by Western blotting. We found that the expression of the cleaved mature form (70 kDa) or the full-length precursor form (120 kDa) of sterol regulatory element binding protein 2 (SREBP2), which is the master transcriptional regulator of cholesterol homeostasis, was not significantly changed in the liver of chow-fed Mc1r LKO mice. However, the ratio between the mature and precursor forms of SREB2 was reduced in Mc1r LKO mice (*Figure 2F and G* and *Figure 2—source data 1*). Consequently, the protein levels of SREBP2 target 3-hydroxy-3-methylglutaryl-CoA reductase (HMGCR) and 7-dehydrocholesterol reductase (DHCR7), which are crucially involved in the biosynthesis of cholesterol (*Prabhu et al., 2014*; *Shi et al., 2022*), were also significantly reduced in Mc1r LKO mice (*Figure 2F and H*).

## Hepatocyte-specific MC1-R deficiency disturbs bile acid metabolism

Based on the previous finding of disturbed bile acid metabolism in global MC1-R deficient mice on *Apoe*[-/-] background (*Rinne et al., 2018*), we were curious to investigate whether the hepatocyte-specific MC1-R knockout model recapitulates this phenotype. To this end, we quantified total and individual bile acids (BA) in the liver, feces, and plasma of chow-fed Mc1r LKO mice by liquid chromatography-mass spectrometry. We found that the total amount of BAs was markedly reduced in the plasma and to some extent (p=0.06) also in the feces of Mc1r LKO mice (*Figure 3B and C*), while the size of hepatic BA pool remained unchanged (*Figure 3A*). These changes were largely attributable to the reduction in secondary BAs (*Figure 3B and C*). Quantification of primary BA species in the plasma revealed that the levels of taurine-conjugated cholic acid (CA) and ursodeoxycholic acid (UDCA) were lower in Mc1r LKO mice. In terms of secondary BAs, Mc1r LKO mice showed significantly reduced plasma levels of taurine-conjugated deoxycholic acid (DCA), hyodeoxycholic acid (HDCA), and $\omega$-muricholic acid ($\omega$-MCA) (*Figure 3E* and *Figure 3—source data 1*). The amount of DCA was also lower in the liver of Mc1r LKO mice (*Figure 3—figure supplement 1* and *Figure 3—figure supplement 1—source data 1*), while in the feces, HDCA, litocholic acid (LCA) and 12-keto litocholic acid (12-oxo LCA), which is the primary metabolite of DCA, were significantly reduced by MC1-R deficiency (*Figure 3—figure supplement 1* and *Figure 3—figure supplement 1—source data 2*). Furthermore, the relative proportions of primary BAs in the plasma indicate that hepatocyte-specific MC1-R deficiency reduced the amount of CA with an accompanying increase in CDCA and UDCA (*Figure 3F*). This BA profile is further reflected as a significant reduction in the plasma ratio of CA:CDCA (*Figure 3G*).

To address possible causes of disturbed BA metabolism in Mc1r LKO mice, we quantified the hepatic expression of genes encoding for BA synthetizing enzymes (*Figure 4A*). Although the expression of cholesterol 7 alpha-hydroxylase (encoded by *Cyp7a1*), which is the first and rate-liming enzyme in BA synthesis (*Chiang, 2004*), was unchanged, Mc1r LKO mice demonstrated significant upregulation of sterol 12α-hydroxylase (encoded by *Cyp8b1*) and sterol 27-hydroxylase (encoded by *Cyp27a1*) (*Figure 4A*). Furthermore, Mc1r LKO mice had reduced mRNA levels of steroidogenic acute regulatory protein 1 (*Stard1*), which facilitates the trafficking of cholesterol to mitochondria and thus feeds the alternative mitochondrial pathway of BA synthesis (*Figure 4A*; *Pandak et al., 2002*; *Ren et al., 2004*). Second, we quantified the hepatic mRNA levels of transporters responsible for the uptake of BAs and their excretion into bile and systemic circulation (*Figure 4B*). We found that the expression of sodium/bile acid cotransporter (encoded by *Ntcp*), which accounts for the majority (~90%) of BA uptake from the portal circulation (*Dawson et al., 2009*), was upregulated in the liver of Mc1r LKO mice, while bile salt export pump (*Bsep*) showed no change at the mRNA level (*Figure 4B*). An alternative basolateral export of BAs is mediated by the heterodimeric organic solute transporter OSTα/OSTβ and the multidrug resistance-associated proteins MRP3 and MRP4 (*Dawson et al., 2009*), the

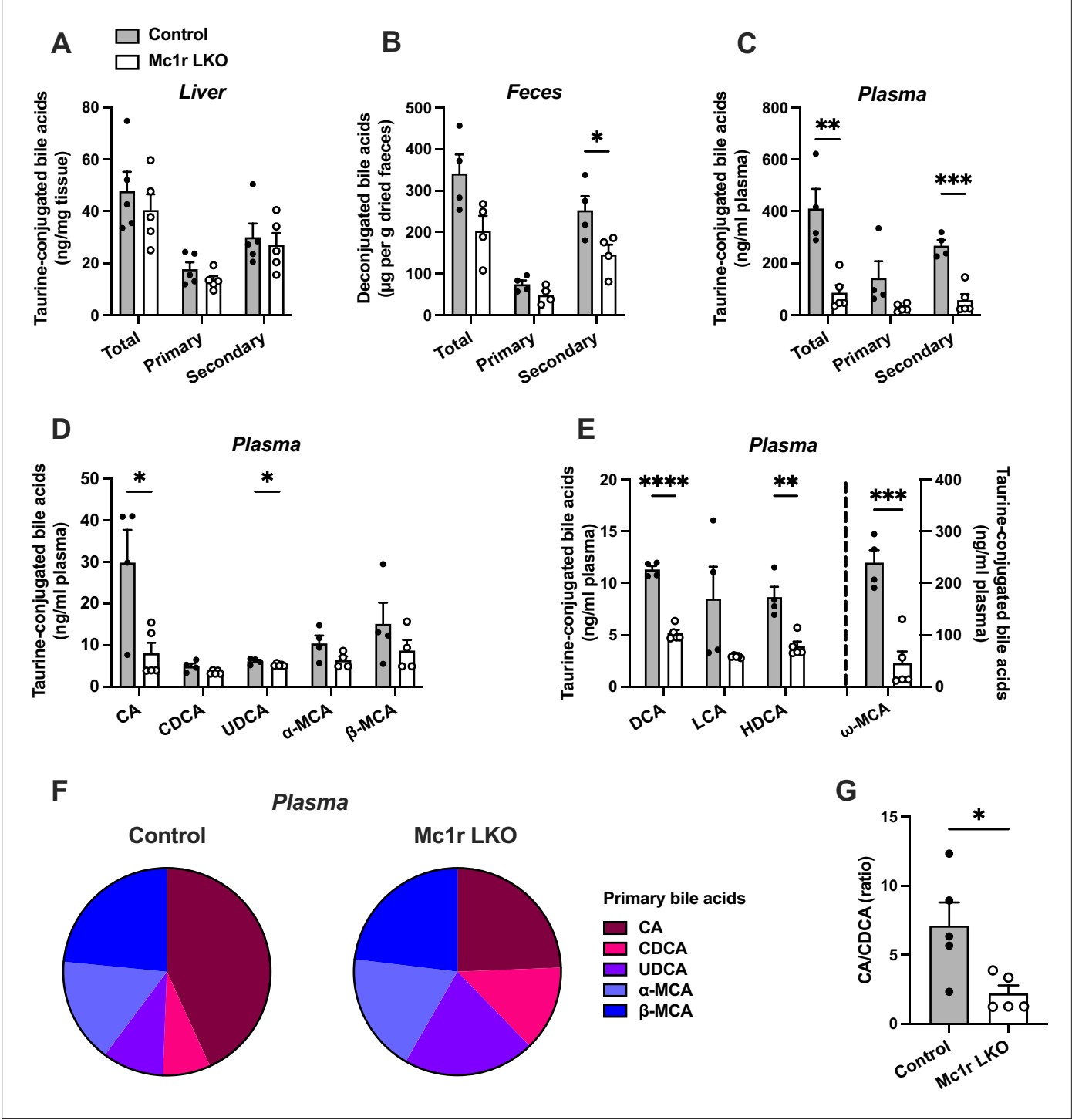

**Figure 3.** Hepatocyte-specific melanocortin 1 receptor (MC1-R) deficiency disturbs bile acid metabolism. (**A–C**) Quantification of total, primary, and secondary bile acids in the liver, plasma and feces of chow-fed control (*Mc1r^fl/+ Alb^Cre/+*), and Mc1r LKO mice. (**D**) Quantification of individual primary bile acids in the plasma of chow-fed control and Mc1r LKO mice. (**E**) Quantification of individual secondary bile acids in the plasma of chow-fed control and Mc1r LKO mice. (**F**) Relative proportions of individual primary bile acids in the plasma of chow-fed control and Mc1r LKO mice. (**G**) The ratio of cholic acid (CA) to chenodeoxycholic acid (CDCA) in the plasma of chow-fed control and Mc1r LKO mice. Values are mean ± SEM, n=4–5 mice per group in each graph. *p<0.05, **p<0.01, ***p<0.001, and ****p<0.0001 versus control mice by Student's t-test. CA indicates cholic acid; CDCA, chenodeoxycholic acid; UDCA, ursodeoxycholic acid; MCA, muricholic acid; DCA, deoxycholic acid; LCA, litocholic acid; HDCA, hyodeoxycholic acid (HDCA).

The online version of this article includes the following source data and figure supplement(s) for figure 3:

*Figure 3 continued on next page*

*Figure 3 continued*

**Source data 1.** Ultra-high performance liquid chromatography–tandem mass spectrometry (UHPLC-MS/MS) analysis of individual bile acids in the plasma.

**Figure supplement 1.** Bile acid profiles in the liver and feces of Mc1r LKO mice.

**Figure supplement 1—source data 1.** Ultra-high performance liquid chromatography–tandem mass spectrometry (UHPLC-MS/MS) analysis of individual bile acids in the liver.

**Figure supplement 1—source data 2.** Ultra-high performance liquid chromatography–tandem mass spectrometry (UHPLC-MS/MS) analysis of individual bile acids in the feces.

last of which was downregulated in Mc1r LKO mice (*Figure 4B*). Third, among different nuclear receptors that regulate the transcription of BA enzymes and transporters, farnesoid X receptor (*Fxr*) and hepatocyte nuclear factor 4α (*Hnf4a*) were significantly upregulated in the liver of Mc1r LKO mice (*Figure 4C*).

Finally, we selected the genes, which were differently expressed in Mc1r LKO mice and could potentially explain the observed BA profile in these mice, and quantified the corresponding protein levels of these gene products in the liver by Western blotting. In good agreement with the mRNA

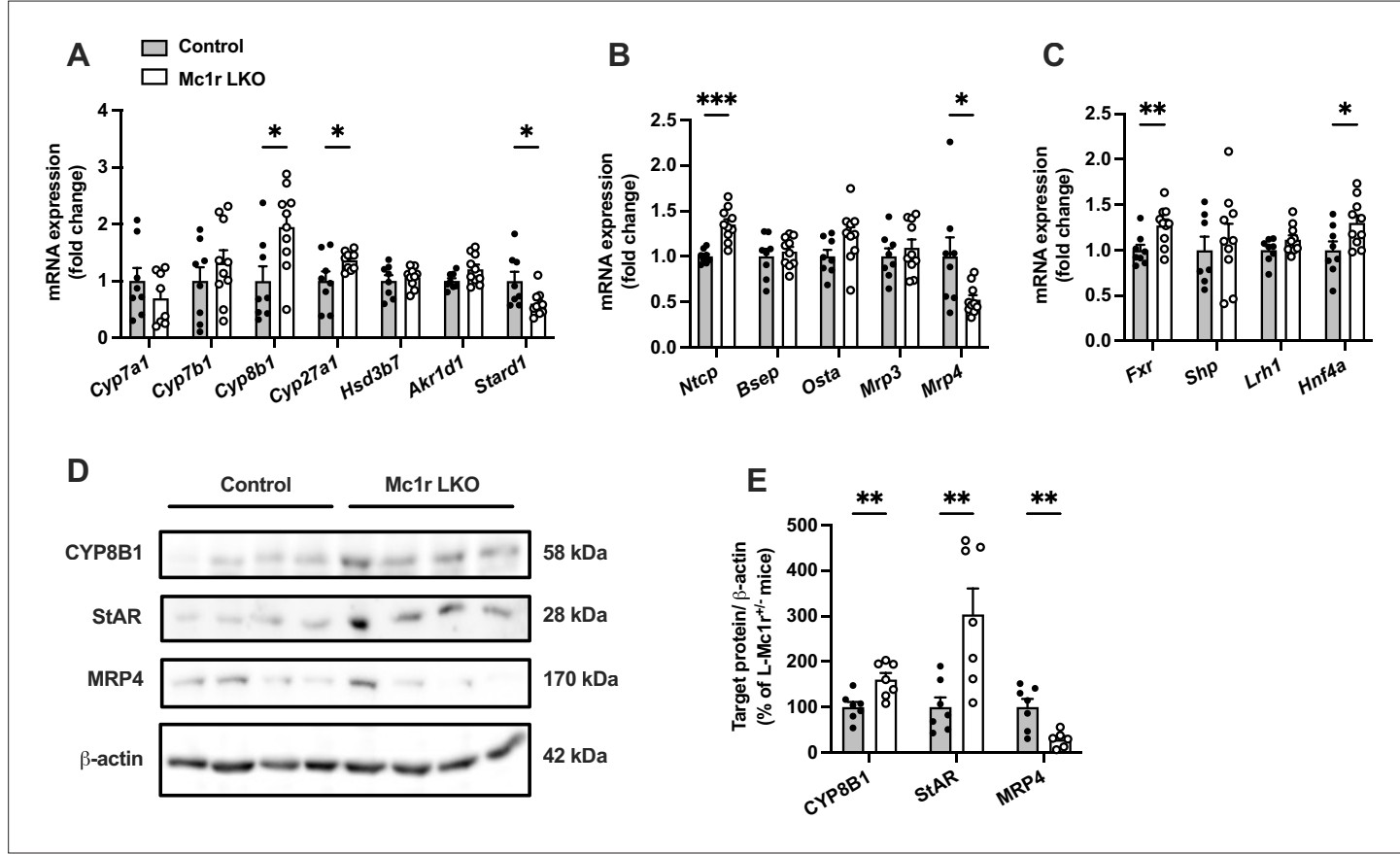

**Figure 4.** Hepatocyte-specific melanocortin 1 receptor (MC1-R) deficiency affects the expression of genes involved in bile acid synthesis and transport. (**A, B**) Quantitative real-time polymerase chain reaction (qPCR) analysis of genes involved in the bile acid synthesis and transport in the liver of chow-fed control (*Mc1r^fl/+ Alb^Cre/+*) and Mc1r LKO mice. (**C**) qPCR analysis of nuclear receptor genes that regulate the transcription of bile acid enzymes and transporters. (**D, E**) Representative Western blots and quantification of CYP8B1, StAR, and MRP4 protein levels in the liver of chow-fed control and Mc1r LKO mice. Values are mean ± SEM, n=7–10 mice per group in each graph. *p<0.05, **p<0.01, and ***p<0.001 versus control mice by Student's t-test. *Cyp7a1*, cholesterol 7 alpha-hydroxylase; *Cyp7b1*, 25-hydroxycholesterol 7-alpha-hydroxylase; *Cyp8b1*, sterol 12-alpha-hydroxylase; *Cyp27a1*, sterol 27-hydroxylase; *Stard1*, steroidogenic acute regulatory protein; *Fxr*, farnesoid X receptor; *Lrh1*, liver receptor homologue 1; *Bsep*, bile-salt export pump; *Ntcp*, Na⁺-taurocholate cotransporting polypeptide; *Hnf4a*, hepatocyte nuclear factor 4 alpha.

The online version of this article includes the following source data for figure 4:

**Source data 1.** Uncropped Western blots for *Figure 4D*.

level changes, Mc1r LKO mice showed higher CYP8B1 and lower MRP4 protein expression compared to control mice (*Figure 4D and E* and *Figure 4—source data 1*). However, StAR (encoded by *Stard1*) protein level was significantly increased in the liver of Mc1r LKO mice (*Figure 4D and E*), which contradicts the mRNA level finding.

## The endogenous MC1-R agonist α-MSH reduces cellular cholesterol content and enhances LDL and HDL uptake in HepG2 cells

The finding of MC1-R expression in the mouse liver and the phenotype of enhanced cholesterol accumulation in Mc1r LKO mice led us to investigate the effects and underlying mechanisms of MC1-R activation in human HepG2 cells. First, we aimed to verify that human hepatocytes also express MC1-R. Indeed, HepG2 cells clearly express MC1-R protein (*Figure 5A* and *Figure 5—source data 1*). Consistent with the finding of reduced MC1-R expression in the liver of Western diet-fed mice, loading of HepG2 cells with palmitic acid (a saturated free fatty acid) caused a rapid decrease of MC1-R protein expression (*Figure 5A* and *Figure 5—source data 1*). However, exposure of HepG2 cells to excess LDL cholesterol (*Figure 5B* and *Figure 5—source data 1*) or treatment with the HMGCR inhibitor atorvastatin (*Figure 5C* and *Figure 5—source data 1*) to lower cellular cholesterol content did not change MC1-R protein level. Second, we studied how MC1-R activation affects cholesterol metabolism in HepG2 cells. For this purpose, HepG2 cells were stimulated with the endogenous MC1-R agonist α-MSH and the amount of cellular free cholesterol was quantified using Filipin staining. We observed that α-MSH (1 µM) significantly decreased the free cholesterol content in HepG2 cells with an effect appearing after 3 hr and plateauing towards the 24 hr time point (*Figure 5D*). In terms of concentration-responsiveness, cholesterol content was already reduced with a subnanomolar concentration (0.1 nM) of α-MSH, and the maximal response was achieved with 1 µM α-MSH (*Figure 5E*). The reduction in cellular cholesterol was accompanied by significant increases in LDL and HDL uptake (*Figure 5F and G*), as evaluated after 24 hr treatment with α-MSH using fluorescently labeled lipoprotein particles (Dil-LDL and Dil-HDL). In good agreement with these findings, gene expression analysis revealed that α-MSH upregulated LDL receptor (*LDLR*) and the HDL receptor SR-BI (*SCARB1*) mRNA levels in a concentration-dependent manner (*Figure 5H,I*). These effects were only apparent at a 3 hr time point, which probably reflects the short half-life of α-MSH (*Redding et al., 1978*; *Rudman et al., 1983*). Nevertheless, upregulated *LDLR* and *SCARB1* mRNA levels translated into more sustained increases in the corresponding protein levels (LDLR and SR-BI) after α-MSH treatment (*Figure 5I and J* and *Figure 5—source data 1*). Protein level analyses further showed that α-MSH had no effect on the expression of the cholesterol biosynthetic enzymes HMGCR and DHCR7 (*Figure 5J and K* and *Figure 5—source data 1*). Intriguingly, we observed that LDLR expression at the cell surface, as quantified by flow cytometry, was markedly increased already after 1 hr of treatment with 1 µM α-MSH (*Figure 5L*). Finally, in terms of BA metabolism, it appeared that α-MSH increased CA concentration in the culture medium of HepG2 cells without any effect on CDCA concentration (*Figure 5—figure supplement 1* and *Figure 5—figure supplement 1—source data 1*). Consequently, the ratio of CA to CDCA was significantly increased in response to α-MSH treatment (*Figure 5—figure supplement 1*). Supporting this finding, Western blotting analysis showed that α-MSH upregulated CYP8B1 (*Figure 5—figure supplement 1* and ), which is the major determinant of CA:CDCA ratio (*Pandak and Kakiyama, 2019*).

## Selective activation of MC1-R mimics the effects of α-MSH in HepG2 cells

Since α-MSH can also bind and activate other MC-R subtypes, we aimed to verify that the effects induced by α-MSH were particularly derived from the activation of MC1-R. To this end, we repeated the key experiments using LD211, which is a highly potent and selective agonist for MC1-R with no detectable binding to other MC-R subtypes (*Doedens et al., 2010*). Closely mirroring the effect observed with α-MSH, selective activation of MC1-R with LD211 led to a concentration-dependent reduction in cellular cholesterol amount (*Figure 6A*). LD211 also significantly increased LDL and HDL uptake (*Figure 6B and C*), which was accompanied by upregulation of *LDLR* and *SCARB1* mRNA expression (*Figure 6D*). In comparison with α-MSH (*Figure 4G and H*), LD211 caused more sustained changes in gene expression (*Figure 6D*), which could be explained by the cyclic and more stable structure of LD211 (*Doedens et al., 2010*). Despite the clear-cut effects observed at the mRNA level, LD211

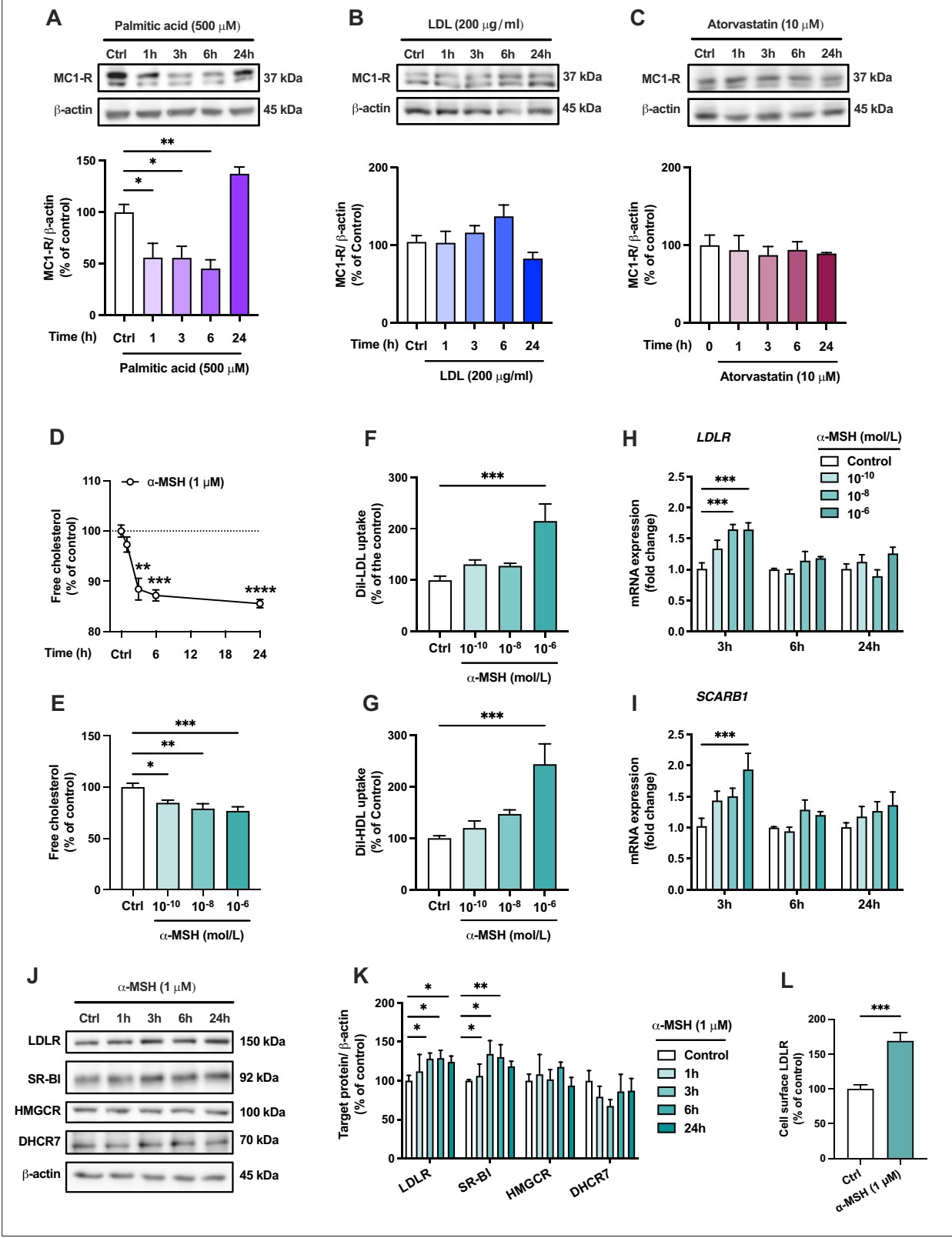

**Figure 5.** The endogenous melanocortin 1 receptor (MC1-R) agonist α-MSH reduces cellular cholesterol content and enhances low-density lipoprotein (LDL) and high-density lipoprotein (HDL) uptake in HepG2 cells. (**A–C**) Representative Western blots and quantification of MC1-R protein level in HepG2 cells treated with palmitic acid (500 μM), LDL (200 μg/ml), or atorvastatin (10 μM) for 1, 3, 6, or 24 hr. (**D**) Quantification of free cholesterol content using filipin staining in HepG2 cells treated with α-MSH (1 μM) for 1, 3, 6, or 24 hr. (**E**) Quantification of free cholesterol content in HepG2 cells treated with

*Figure 5 continued on next page*

Figure 5 continued

different concentrations of α-MSH (0.1 nM, 10 nM, or 1 µM) for 24 hr. (**F, G**) Quantification of LDL and HDL uptake in HepG2 cells treated with different concentrations (0.1 nM, 10 nM, or 1 µM) of α-MSH for 24 hr. (**H, I**) Quantitative real-time polymerase chain reaction (qPCR) analysis of LDL receptor (*LDLR*) and *SCARB1* expression in HepG2 cells treated with different concentrations of α-MSH for 3, 6, or 24 hr. (**J, K**) Representative Western blots and quantification of LDL-R and SR-BI proteins levels in HepG2 cells treated with 1 µM α-MSH for 1, 3, 6, or 24 hr. (**L**) Quantification of cell surface LDLR by flow cytometry in HepG2 cells treated with 1 µM α-MSH for 24 hr. Values are mean ± SEM, n=3–6 per group in each graph. *p<0.05 and **p<*0.01* for the indicated comparisons by one-way ANOVA and Dunnet *post hoc* tests (**A–K**) or by Student's t-test (**L**).

The online version of this article includes the following source data and figure supplement(s) for figure 5:

**Source data 1.** Uncropped Western blots for *Figure 5A, B, C and J*.

**Figure supplement 1.** The effects of α-MSH on bile acid production in HepG2 cells.

**Figure supplement 1—source data 1.** Ultra-high performance liquid chromatography–tandem mass spectrometry (UHPLC-MS/MS) analysis of individual bile acids in the cell culture medium.

**Figure supplement 1—source data 2.** Uncropped Western blots for *Figure 5—figure supplement 1*.

did not significantly affect LDLR and SR-BI protein levels, as detected by Western blotting (*Figure 6E* and *Figure 6—source data 1*). However, the cell surface expression of LDLR, which is a major determinant of LDL uptake rate, was markedly increased in HepG2 cells after 3-, 6-, and 24 hr treatment with LD211 (*Figure 6F*). Finally, the subnanomolar concentration of LD211 also down-regulated the fibrosis-associated genes *TGFB1*, *ACTA2,* and *COL1A1* in HepG2 cells (*Figure 6—figure supplement 1*). α-MSH appeared to also downregulate the expression of the fibrotic genes *TGFB1* and *COL1A1* (*Figure 6—figure supplement 1*), but these changes did not reach statistical significance (p=0.08 and 0.051, respectively). Taken together, these results demonstrate that the α-MSH-induced effects were largely reproducible by selective MC1-R activation with LD211.

## MC1-R activation engages multiple signaling mechanisms to regulate cholesterol metabolism in HepG2 cells

We next aimed to investigate intracellular signaling cascades that might be activated in response to MC1-R stimulation. Since most MC-Rs are known to be coupled to Gs proteins, we first measured intracellular cAMP levels in α-MSH-treated HepG2 cells. However, α-MSH did not either increase or decrease cAMP levels, while the adenylyl cyclase activator forskolin, as a positive control, induced a robust increase in cAMP level (*Figure 7A*). We also screened other potential signaling pathways of MC1-R and found that mitogen-activated protein kinases, ERK (extracellular-signal-regulated kinase) and JNK (c-Jun N-terminal kinase), are affected by α-MSH treatment. The highest concentration of α-MSH (1 µM) reduced ERK phosphorylation (p-ERK1/2) at 5- and 15 min time points (*Figure 7—figure supplement 1*). Profiling of the concentration-response however revealed that the phosphorylation levels of ERK and JNK are most significantly reduced at the lowest tested concentration (0.1 nM) of α-MSH and the effects tend to fade away towards higher concentrations (*Figure 7B and C*). In addition, α-MSH induced a rapid phosphorylation (at 5 min) of AMP-activated protein kinase (AMPK) (*Figure 7D* and *Figure 7—source data 1*), and the maximal effect was observed with 1 µM concentration (*Figure 7—figure supplement 1* and *Figure 7—figure supplement 1—source data 1*), indicating a more conventional concentration-response. Finally, we tested whether the α-MSH-induced reduction in cellular cholesterol content is mediated by AMPK phosphorylation. Interestingly, the AMPK inhibitor dorsomorphin increased cholesterol content and reversed the concentration-response to α-MSH (*Figure 7E*). In the presence of dorsomorphin, the lowest concentration of α-MSH showed the strongest response, while the effect of 1 µM α-MSH was completely blocked (*Figure 7E*).

To verify the dependence of the observed effects on MC1-R activation, we repeated the signaling experiments using LD211 as a selective MC1-R agonist. LD211 had no effect on cAMP level (*Figure 7—figure supplement 2*) but reduced the phosphorylation level of ERK1/2 and JNK (*Figure 7—figure supplement 2*) and induced phosphorylation of AMPK (*Figure 7—figure supplement 2* and *Figure 7—figure supplement 2—source data 1*), thus closely matching the phenotype of α-MSH-treated cells. Furthermore, we observed that AMPK inhibition with dorsomorphin completely abolished the cholesterol-lowering effect of LD211 (*Figure 7—figure supplement 2*). Collectively, the results demonstrate that α-MSH evokes multiple signaling pathways and that the effects of α-MSH on cholesterol metabolism are not reliant on one single pathway.

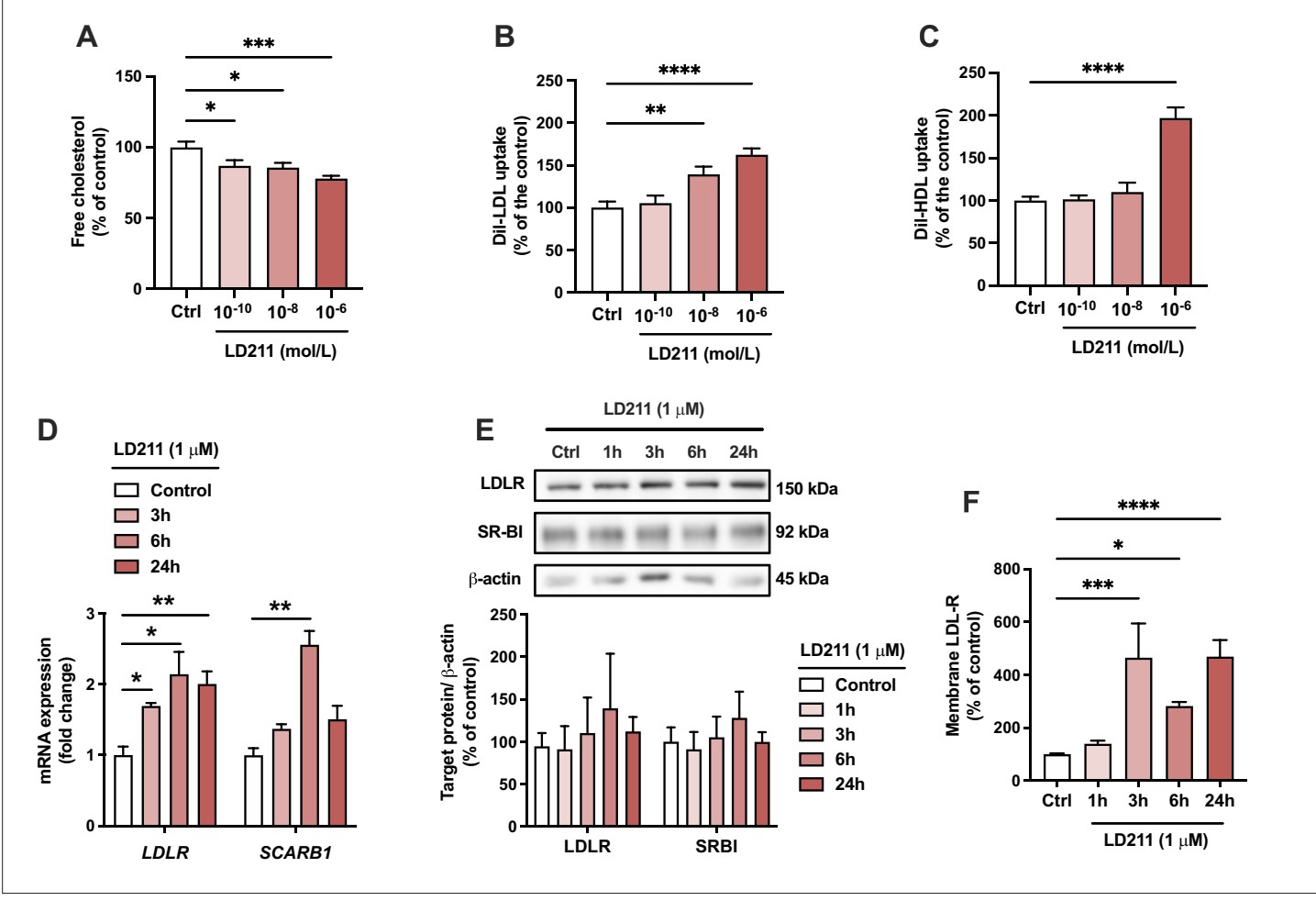

**Figure 6.** Selective activation of melanocortin 1 receptor (MC1-R) mimics the actions of α-MSH in HepG2 cells. (**A**) Quantification of free cholesterol content using filipin staining in HepG2 cells treated with different concentrations of the selective MC1-R agonist LD211 (0.1 nM, 10 nM, or 1 μM) for 24 hr. (**B, C**) Quantification of low-density lipoprotein (LDL) and high-density lipoprotein (HDL) uptake in HepG2 cells treated with different concentrations (0.1 nM, 10 nM, or 1 μM) of LD211 for 24 hr. (**D**) Quantitative real-time polymerase chain reaction (qPCR) analysis of LDL receptor (*LDLR)* and *SCARB1* expression in HepG2 cells treated with 1 μM LD211 for 3, 6, or 24 hr. (**E**) Representative Western blots and quantification of LDL-R and SR-BI proteins levels in HepG2 cells treated with 1 μM LD211 for 1, 3, 6, or 24 hr. (**F**) Quantification of cell surface LDLR by flow cytometry in HepG2 cells treated with 1 μM LD211 for 24 hr. Values are mean ± SEM, n=3–6 per group in each graph. *p<0.05, **p<0.01, ***p<0.001, and ****p<0.0001 for the indicated comparisons by one-way ANOVA and Dunnet *post hoc* tests.

The online version of this article includes the following source data and figure supplement(s) for figure 6:

**Source data 1.** Uncropped Western blots for *Figure 6E*.

**Figure supplement 1.** The effects of α-MSH and the selective melanocortin 1 receptor (MC1-R) agonist LD211 on the expression of pro-inflammatory and fibrotic genes in HepG2 cells.

## Discussion

In the present study, we investigated the role of MC1-R in hepatocytes and its possible involvement in cholesterol and bile acid metabolism. First, extending previous observations on the hepatic expression of *Mc1r* mRNA, we show that MC1-R protein is widely present in the mouse liver and downregulated in response to feeding a cholesterol-rich diet. Second, hepatocyte-specific MC1-R deficiency rendered mice susceptible to enhanced accumulation of cholesterol and triglycerides in the plasma and liver (*Figure 8*). Loss of MC1-R signaling in hepatocytes disturbed also bile acid metabolism. Third, in vitro experiments using HepG2 cells revealed that triggering MC1-R signaling either with the endogenous ligand α-MSH or the selective MC1-R agonist LD211 reduced cellular cholesterol content

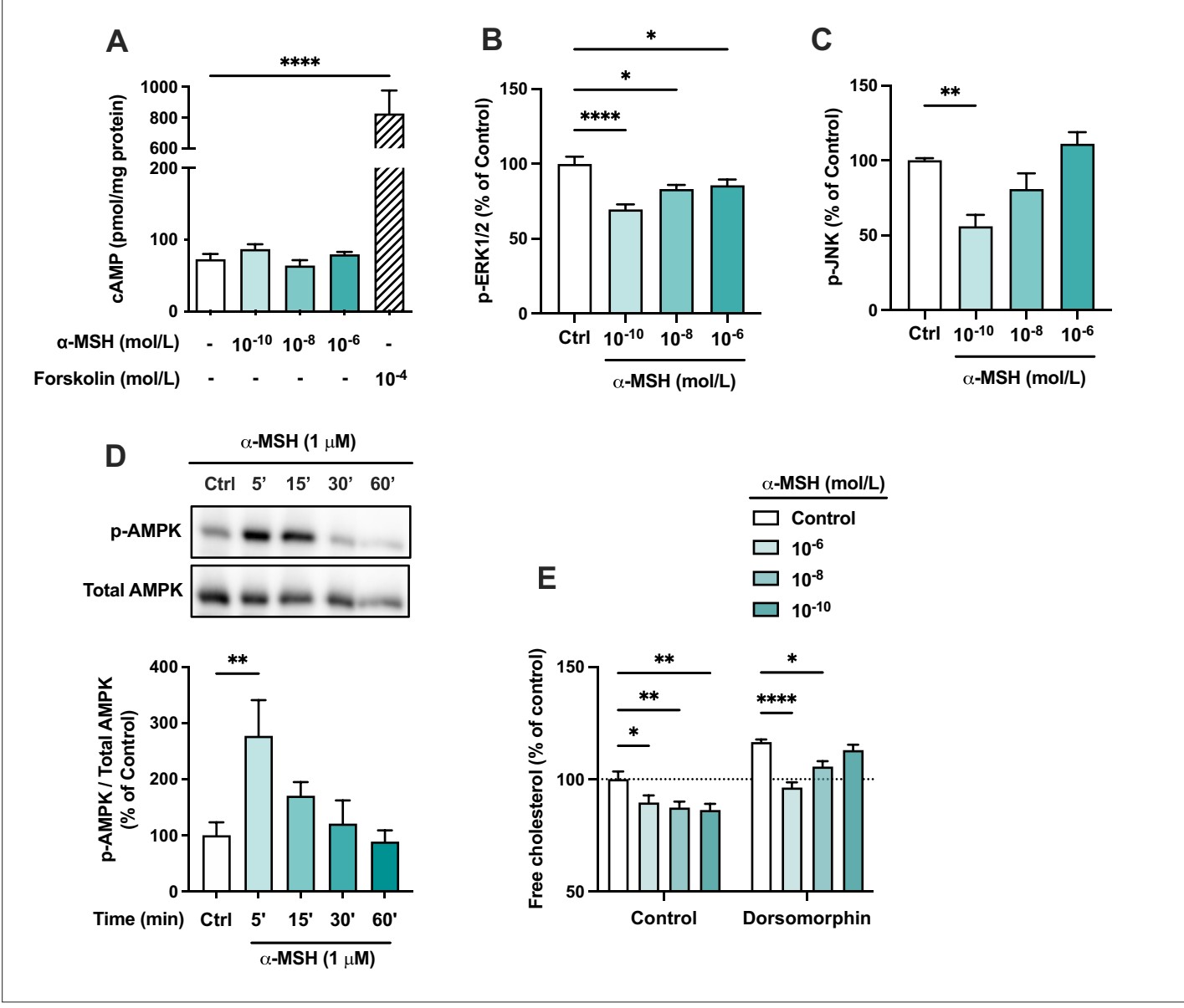

**Figure 7.** The effects of α-MSH on intracellular signaling pathways in HepG2 cells. (**A**) Quantification of intracellular cAMP level in HepG2 cells treated with different concentrations of α-MSH (0.1 nM, 10 nM, or 1 μM) for 30 min. The adenylyl cyclase activator forskolin (10 μM) was used as a positive control. (**B, C**) Quantification of phosphorylated ERK1/2 and JNK by ELISA assays in HepG2 cells treated with different concentrations of α-MSH (0.1 nM, 10 nM, or 1 μM) for 10 min. (**D**) Representative Western blots and quantification of phosphorylated AMPK level (p-AMPK normalized against total AMPK) in HepG2 cells treated with 1 μM α-MSH for 5, 15, 30, or 60 min. (**E**) Quantification of free cholesterol content using filipin staining in HepG2 cells treated with different concentrations of α-MSH (0.1 nM, 10 nM, or 1 μM) for 24 hr in the presence or absence of the AMP-activated protein kinase (AMPK) inhibitor dorsomorphin (1 μM). Values are mean ± SEM, n=3–6 per group in each graph. *p<0.05, **p<0.01, and ****p<0.0001 for the indicated comparisons by one-way ANOVA and Dunnet *post hoc* tests.

The online version of this article includes the following source data and figure supplement(s) for figure 7:

**Source data 1.** Uncropped Western blots for *Figure 7D*.

**Figure supplement 1.** The effects of α-MSH on the phosphorylation of ERK1/2 and AMP-activated protein kinase (AMPK) in HepG2 cells.

**Figure supplement 1—source data 1.** Uncropped Western blots for *Figure 7—figure supplement 1*.

**Figure supplement 2.** The effects of the selective melanocortin 1 receptor (MC1-R) agonist LD211 on intracellular signaling pathways in HepG2 cells.

**Figure supplement 2—source data 1.** Uncropped Western blots for *Figure 7—figure supplement 2*.

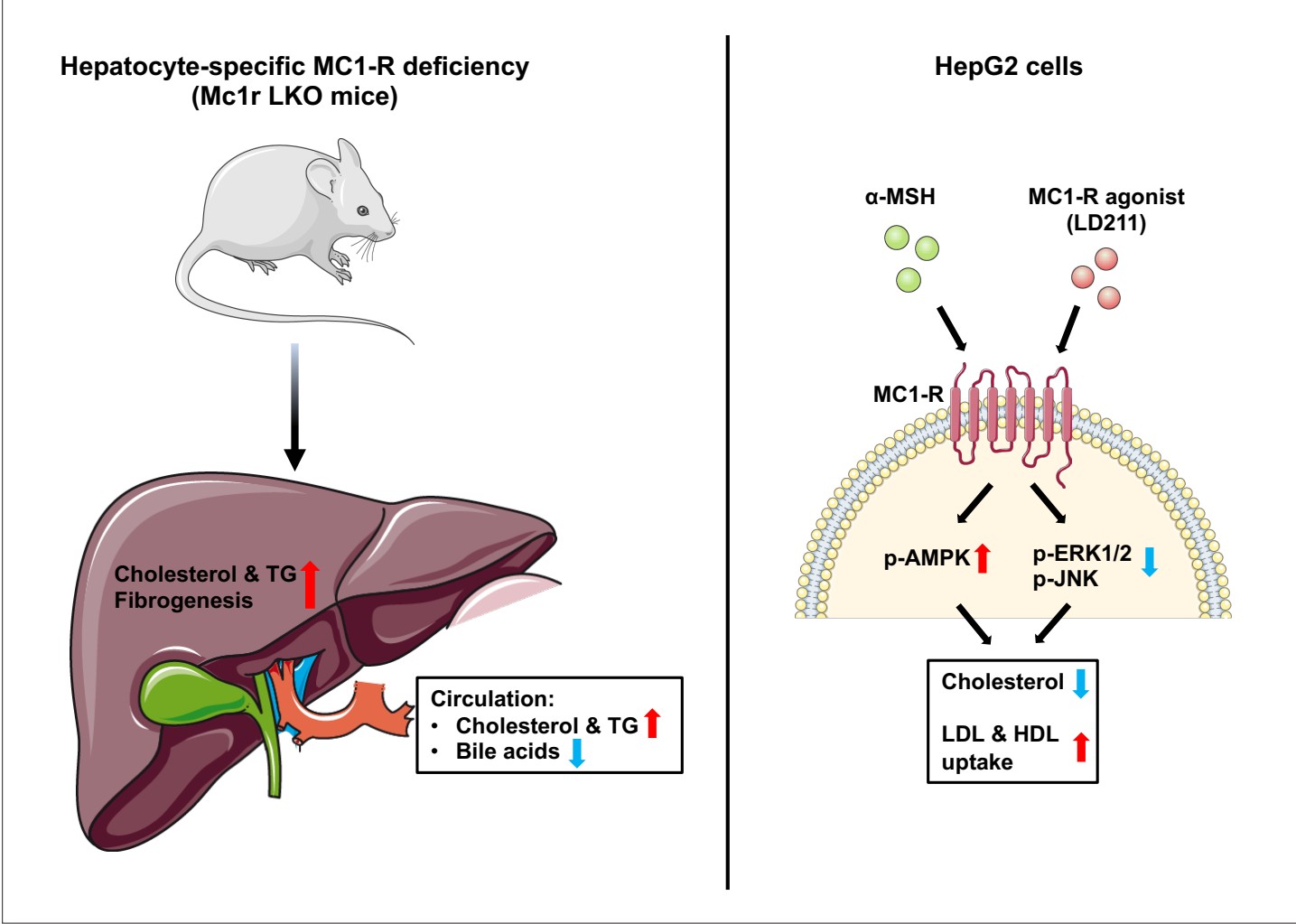

**Figure 8.** Schematic figure illustrates the role of melanocortin 1 receptor (MC1-R) in regulating cholesterol and bile acid homeostasis. Hepatocyte-specific MC1-R deficiency enhanced the accumulation of cholesterol and triglycerides (TG) in the liver, promoted fibrogenesis, and lead to a disturbance in bile acid metabolism. Conversely, activation of MC1-R with the endogenous agonist α-MSH or the synthetic agonist LD211 reduced cellular cholesterol levels and increased the uptake of low-density lipoprotein (LDL) and high-density lipoprotein (HDL) particles in cultured HepG2 cells. p-AMPK indicates phosphorylated AMP-activated protein kinase; p-ERK1/2, phosphorylated extracellular-signal-regulated kinase 1/2; p-JNK, phosphorylated c-Jun N-terminal kinase. The figure was partly generated using Servier Medical Art, provided by Servier, licensed under a Creative Commons Attribution 3.0 unported license.

and enhanced the uptake of HDL and LDL cholesterol (*Figure 8*), which are preventive mechanisms against hypercholesteremia and associated cardiovascular complications.

Previous findings of *MC1R* mRNA expression in the rat and human liver (*Gatti et al., 2006*; *López et al., 2007*; *Malik et al., 2012*) and that global MC1-R deficiency aggravated hypercholesterolemia in Apoe$^{-/-}$ mice (*Rinne et al., 2018*) led us to hypothesize that MC1-R regulates cholesterol metabolism in hepatocytes. *Mc1r* mRNA expression in the rat liver was previously found to increase after turpentine oil-induced acute phase response (*Malik et al., 2012*), while *MC1R* was downregulated in liver biopsies from brain-dead organ donors (*Gatti et al., 2006*), suggesting that inflammatory processes might modulate hepatic MC1-R expression. Here, we show that the MC1-R is present also in the mouse liver where it localizes mainly in hepatocytes but also in other cell types such as cholangiocytes. Of note, hepatic MC1-R mRNA and protein levels were reduced after feeding mice a cholesterol-rich Western diet. Consistently, hepatic *MC1R* expression was downregulated in patients with NAFLD or NASH. We also analyzed MC1-R expression in HepG2 cells to gain further insight into the regulation of MC1-R expression in hepatocytes. Intriguingly, acute changes in cellular cholesterol load, as produced by treatment with atorvastatin or LDL-cholesterol, did not affect the MC1-R protein

level, while treatment with palmitic acid evoked an immediate reduction in MC1-R expression. These findings suggest that cholesterol per se does not regulate MC1-R expression but it might be rather modulated by inflammatory processes that are triggered by lipid overload.

To investigate the regulatory role of hepatic MC1-R, we generated Mc1r LKO mice and found that hepatocyte-specific MC1-R deficiency induced hypercholesteremia and higher liver weight, which was accompanied by increased total cholesterol and triglyceride levels in the liver. Mc1r LKO mice closely phenocopied the features of *Apoe*$^{-/-}$ mice with global deficiency of MC1-R (*Rinne et al., 2018*), which displayed enhanced hypercholesteremia and cholesterol accumulation in the liver. Thus, the present findings suggest that the disturbed cholesterol metabolism previously observed in global MC1-R deficient mice was attributable to the loss of MC1-R signaling in hepatocytes. The precise mechanism leading to enhanced hepatic lipid accumulation and hypercholesterolemia in Mc1r LKO mice is not yet clear. Under normal physiological conditions, SREBP1c preferentially activates the genes of fatty acid and triglyceride biosynthesis pathway, while SREBP2 is considered as the master regulator of cholesterol metabolism (*Hua et al., 1993*; *Yokoyama et al., 1993*). In the presence of excess cellular cholesterol, transcriptional induction and posttranslational activation of SREBP2 are suppressed, which, in turn, downregulates *Hmgcr* and *Dhcr7* and reduces cholesterol synthesis as a counterregulatory mechanism. Therefore, given the increase in hepatic cholesterol content, it was expected that HMGCR and DHCR7 protein levels were reduced in the liver of Mc1r LKO mice. These data, in combination with the finding that MC1-R activation did not affect HMGCR or DHCR7 expression in HepG2 cells, could indicate that hepatic MC1-R signaling does not directly control cholesterol synthesis. Although the phenotype observed in Mc1r LKO mice might be, at least partly, caused by disturbed cholesterol conversion into bile acids, further research is warranted to dissect the exact mechanism by which hepatic MC-1R deficiency triggers hypercholesterolemia and enhanced lipid accumulation in the liver.

A central feature of the Mc1r LKO mouse model was liver steatosis that occurred on a normal chow diet and without any signs of increased susceptibility for obesity. Although the precise mechanisms leading to NAFLD remain unclear, human and animal studies have shown that under normal caloric intake, the development of NAFLD is strongly associated with disturbance in liver cholesterol metabolism (*Kainuma et al., 2006*; *Min et al., 2012*; *Simonen et al., 2011*). Hence, the increased levels of plasma and liver cholesterol in Mc1r LKO mice may have secondarily led to the accumulation of triglycerides. Simple steatosis can progress to NASH that is characterized by progressive inflammation, oxidative stress, and fibrosis. Mc1r LKO mice showed enhanced liver fibrosis without any signs of elevated inflammation. Persistent lipid overload and consequent lipotoxicity could be the cause of fibrosis in Mc1r LKO mice. However, MC1-R signaling has been shown to mediate anti-fibrotic effects and protect against skin fibrosis and systemic sclerosis (*Böhm and Stegemann, 2014*; *Kondo et al., 2022*), which opens the possibility that hepatocyte-specific MC1-R deficiency directly induces fibrogenesis. This view is further supported by the finding that selective MC1-R activation down-regulated fibrotic genes in HepG2 cells.

Mc1r LKO mice also displayed a unique bile acid profile with reduced secondary bile acid levels, particularly in the plasma and feces. Furthermore, the ratio of CA to CDCA was reduced in the plasma of Mc1r LKO mice. This phenotype recapitulates some of the key features observed in mice with global deficiency of MC1-R (*Rinne et al., 2018*). Specifically, the relative amount of CA and the ratio of CA to CDCA were reduced in both mouse models. In the quest for a possible explanation for this phenotype, we found that CYP8B1 was upregulated both at the mRNA and protein level in the liver of Mc1r LKO mice. This contradicts the finding of a reduced CA:CDCA ratio, which is predominantly determined by the enzymatic activity of CYP8B1 that is required for CA synthesis (*Chiang, 2004*). Therefore, it is plausible that BA synthesis *via* the classical pathway is disturbed in Mc1r LKO mice due to dysfunctional CYP8B1, which leads to a compensatory enhancement of BA synthesis *via* the alternative pathway. This could explain the upregulation of StAR and CYP27A1, which operate in the mitochondria to feed the alternative BA pathway. Under physiological conditions, the majority of BAs are produced by the classical pathway, while in liver diseases such as NAFLD, the alternative BA pathway may become more dominant when compensating disturbances in the classical BA synthesis pathway (*Chiang, 2004*; *Crosignani et al., 2007*; *Lake et al., 2013*). Patients with NASH and NASH-driven hepatocellular carcinoma have also been reported to have increased hepatic StAR expression (*Caballero et al., 2009*; *Conde de la Rosa et al., 2021*).

In terms of BA transport, the upregulation of NTCP and downregulation of MRP4 in Mc1r LKO mice might also indicate compensatory changes to disrupted BA synthesis. Increased NTCP is likely to enhance BA uptake from the portal circulation and thus enterohepatic circulation of BAs, while reduced MRP4 expression prevents excessive spillover of BAs into the systemic circulation. These changes synergistically help to maintain the liver BA pool in the presence of disturbed BA synthesis. Reduced MRP4 expression in Mc1r LKO mice also provides a mechanistic explanation for the finding of reduced BA levels in the plasma. It remains, however, to be determined whether the dysregulated BA metabolism contributes to the hypercholesteremia and increased hepatic lipid accumulation in Mc1r LKO mice. For instance, CYP7A1 deficiency in both humans and mice causes hypercholesterolemic phenotype with increased hepatic cholesterol content (*Erickson et al., 2003*; *Pullinger et al., 2002*). Because there is a reciprocal interaction between fatty liver disease and dysregulated BA metabolism, it is difficult to determine whether MC1-R deficiency per se disturbs BA metabolism or whether enhanced lipid accumulation in Mc1r LKO mice causes a defect in BA metabolism. However, in vitro experiments with HepG2 cells support the notion that hepatic MC1-R signaling might directly affect BA metabolism, since treatment with the endogenous MC1-R agonist α-MSH modestly increased CYP8B1 expression, CA synthesis, and CA:CDCA ratio. Furthermore, considering that MC1-R was expressed also in cholangiocytes and that Cre recombinase is known to be active in cholangiocytes of *Alb^Cre* transgenic mice (*Lemaigre, 2015*), it is plausible that deficiency of MC1-R signaling in cholangiocytes contributes to the defect of BA metabolism in Mc1r LKO mice. Cholangiocytes in co-operation with hepatocytes are responsible for bile formation and secretion but these cells also significantly contribute to bile modification by absorbing bile acids and other molecules from the biliary tree and returning them to the liver sinusoids (*Tabibian et al., 2013*).

In good agreement with our findings in Mc1r LKO mice, in vitro experiments using HepG2 cells demonstrated that triggering MC1-R signaling with the endogenous agonist α-MSH or the synthetic agonist LD211 induced a reverse phenotype, namely reduction in cellular cholesterol content. MC1-R activation was also associated with enhanced LDL and HDL uptake in HepG2 cells. Inhibition of cholesterol synthesis (e.g. by statins) is known to similarly reduce cellular cholesterol levels in hepatocytes, which, in turn, upregulates LDL-R and reduces plasma total and LDL cholesterol. In the case of MC1-R activation, the increase in LDL-R expression and LDL uptake might be partly independent of the effect on cellular cholesterol content, since the upregulation of LDL-R occurred rapidly after α-MSH treatment (at 1 hr time point) and before any noticeable change in cellular cholesterol level. Therefore, the data suggest that the upregulation of LDL-R, as well as SR-BI, is attributable mainly to direct transcriptional induction by MC1-R signaling. Supporting the therapeutic relevance of this finding, we had previously observed that chronic treatment of atherosclerotic mice with a selective MC1-R agonist increased LDL-R expression in the liver and reduced plasma total cholesterol concentration (*Rinne et al., 2017*). The underlying mechanism remained obscure in that study, but the present findings suggest that the upregulation of LDL-R expression was a consequence of MC1-R activation in hepatocytes. In the current study, we also found that MC1-R activation reduced cellular cholesterol content in HepG2 cells but we were unable to pinpoint the exact molecular-level mechanism for this effect. Since we did not observe any change in the expression of the major cholesterol biosynthetic enzymes (HMGCR or DHCR7), we speculate that MC1-R signaling might inhibit other enzymes in the cholesterol biosynthesis pathway or modulate the phosphorylation state of HMGCR that determines its catalytic activity (*Burg and Espenshade, 2011*). Alternatively, or in addition to other mechanisms, enhanced cholesterol turnover into BAs, as evidenced by increased CA production and CYP8B1 expression in α-MSH-treated HepG2 cells, might reduce cellular cholesterol content. In any case, these data uncover a functional role for MC1-R signaling in hepatic cholesterol metabolism that might be of therapeutic relevance in the management of hypercholesterolemia.

In terms of intracellular signaling, MC1-R activation in HepG2 cells evoked phosphorylation of AMPK and inhibition of ERK1/2 and JNK without any effect on cAMP levels. Melanocortin receptors are classically coupled to Gs protein and cAMP-dependent signaling (*Rodrigues et al., 2015*), but the present findings demonstrate that the MC1-R-mediated effects on cholesterol metabolism occur in a cAMP-independent manner. Mechanistic experiments further revealed that the reduction of cellular cholesterol level by α-MSH was partially reversed after AMPK inhibition with dorsomorphine. However, in the presence of dorsomorphine, low concentrations of α-MSH were still effective in reducing cellular cholesterol content and the concentration-response closely mirrored the profiles

observed in p-JNK and p-ERK1/2 levels in α-MSH-treated HepG2 cells (i.e. U-shaped concentration-response). These results suggest that the α-MSH-mediated reduction in cellular cholesterol content relies on multiple pathways involving AMPK and MAPK signaling pathways. On the other hand, the cholesterol-lowering effect of the selective MC1-R agonist LD211 appeared to be completely dependent on AMPK phosphorylation, which might indicate that this synthetic agonist has a stronger signaling bias toward the AMPK pathway compared to α-MSH. Early in vitro studies have established that AMPK activation reduces cholesterol synthesis by inducing an inactivating phosphorylation of HMGCR (*Clarke and Hardie, 1990*; *Steinberg and Kemp, 2009*), while inhibition of AMPK increases cholesterol synthesis and cholesterol accumulation in the liver (*Loh et al., 2018*). The link between MAPK signaling and cholesterol metabolism has not been widely studied but in vitro experiments using HepG2 cells have demonstrated that ERK1/2 activation phosphorylates SREBP2 (*Kotzka et al., 2004*), which, in turn, is likely to induce e.g., *HMGCR* transcription. The role of hepatic JNK signaling is even less clear in this regard, but some evidence demonstrates that it contributes to diet-induced obesity and hepatic steatosis as well as to cholesterol and BA metabolism (*Manieri et al., 2020*; *Manieri and Sabio, 2015*; *Vernia et al., 2014*). Against this background, it is plausible that both AMPK phosphorylation and inhibition of ERK1/2 and JNK signaling are involved in mediating the effects of α-MSH on cholesterol metabolism in hepatocytes. However, further experiments are warranted to dissect the exact signaling mechanism(s) of MC1-R activation that regulate cholesterol metabolism, e.g., to determine which G-protein subtype is involved in this regulation.

In conclusion, our study uncovers a novel role for MC1-R signaling in hepatic cholesterol and BA metabolism. Hepatocyte-specific MC1-R deficiency increased plasma cholesterol and TG concentration, disturbed BA metabolism, and led to signs of hepatic steatosis and fibrosis. Conversely, MC1-R activation in hepatocytes reduced cellular cholesterol content and increased LDL and HDL uptake, which are preventive mechanisms against hypercholesterolemia and the progression of NAFLD.

## Materials and methods
### Mice
All experiments were performed on adult (3–6 months) female mice. Mice were housed in groups of littermates on a 12 hr light/dark cycle. The numbers of mice studied in each experiment are given in the figure legends. Sample sizes were empirically determined based on previous experience with the experimental models. Where possible, experiments were conducted and analyzed by blinded researchers. Mice were maintained on a regular chow diet (# 2916 C, Teklad Global diet, Envigo) for the entire experimental duration unless otherwise stated. In each experiment, mice were euthanized *via* $CO_2$ asphyxiation, blood was withdrawn, and whole liver was excised and weighed. The experiments were approved by the local ethics committee (Animal Experiment Board in Finland, License Numbers: ESAVI/6280/04.10.07.2016 and ESAVI/1260/2020) and conducted in accordance with the institutional and national guidelines for the care and use of laboratory animals.

Eight-week-old C57Bl/6 J mice (Janvier Labs, France) were fed a regular chow diet or Western-type diet (RD Western Diet, D12079B, Research Diets Inc, NJ, USA) for 12 weeks and used for the quantification of *Mc1r* mRNA and protein levels in the liver. In addition, hepatocyte-specific MC1-R knock-out mice (Mc1r LKO) were generated by breeding mice homozygous for a floxed *Mc1r* allele (*Mc1r^fl/fl* mice, the Jackson Laboratory, strain #029239) (*Takeo et al., 2016*) with transgenic *Alb^Cre/+* mice (B6N.Cg-Speer6-ps1Tg(Alb-cre)21Mgn/J, the Jackson Laboratory, strain #018961) (*Postic et al., 1999*). Age-matched *Mc1r^fl/fl* and *Mc1r^fl/+ Alb^Cre/+* mice were used as controls. Mice were weighed once a week during a monitoring period of 8 weeks (from 8 to 16 weeks of age). A separate cohort of 8-week-old *Mc1r^fl/fl* and Mc1r LKO mice were put on a Western-type diet (RD Western Diet, D12079B) for 12 weeks. Body composition was determined at the start and end of the 8- or 12 week monitoring period by quantitative nuclear magnetic resonance (NMR) scanning (EchoMRI-700, Echo Medical Systems, Houston, TX, USA). At the end of the experiment, genomic DNA samples from the liver were genotyped for the recombined allele using the following primers: ACC ACT GCG TGC TAT CCT G (*Mc1r* 5'forward), ACC CCT TCC CTT GAG GAG T (*Mc1r* 5' reverse), and GAA CTC TGA GGT CAC TAT TTT CTG GAG A (*Mc1r* 3' reverse).

## RNA sequencing data analysis

RNA-sequencing data deposited in the Gene Expression Omnibus (GSE120064) was used to study *MC1R* expression in human liver biopsy samples from patients diagnosed with NAFLD (n=51) or NASH (n=155) and healthy obese control cases (n=10) without any biochemical or histological evidence or NAFLD (*Govaere et al., 2020*). Raw data were processed using FIMM-RNAseq data analysis workflow (version 2.0.7) (*Kangas et al., 2022*). Raw reads were preprocessed using Trim Galore (version 0.6.6). Preprocessed reads were aligned to the reference genome GRCh39 (release 93) from Ensembl. Gene count data was produced using Subreads (version 2.0.1) (*Liao et al., 2013*). Downstream data analysis was performed using R (version 4.2.0; https://www.r-project.org/) package edgeR (version 3.40.2) (*Robinson et al., 2010*). Low gene counts were removed using edgeR's default parameters. Gene counts were normalized using the trimmed mean of M values (TMM) method and expressed as log2 RPKM (reads per kilobase of exon per million reads mapped) values.

## Cell culture

The HepG2 cell line was purchased from ATCC (American Type Culture Collection, Rockville, MD, USA; HB-8065; authenticated by STR profiling; mycoplasma contamination not detected) and maintained in DMEM (Dulbecco's modified Eagle's medium; Sigma-Aldrich) supplemented with 10% (v/v) heat-inactivated fetal bovine serum (FBS; Gibco), 100 U/ml penicillin (Gibco), 100 µg/ml streptomycin (Gibco) at 37 °C in a humid atmosphere with 5% $CO_2$. To study the regulation of MC1-R expression, HepG2 cells were serum-deprived (0.5% FBS) for 16 h and thereafter treated with 200 µg/ml LDL (CliniSciences), 10 µM atorvastatin (Sigma-Aldrich) or 500 µM palmitic acid (Sigma-Aldrich) for 1, 3, 6, or 24 hr. To study the effects of melanocortin system activation, cells were seeded on 12- or 24-well plates and treated with the non-selective MC-R agonist α-MSH (abcam, # ab120189) or the selective MC1-R agonist LD211 (*Doedens et al., 2010*) (compound 28 in the original publication), as indicated in the figure legends.

## Histology, immunohistochemistry, and immunofluorescence staining

A transverse piece of the left lobe was fixed in 10% formalin overnight followed by embedding in paraffin. Four µm-thick serial sections were stained with hematoxylin and eosin (H&E), Picrosirius Red (abcam, # ab150681), or used for MC1-R immunohistochemistry as previously described (*Rinne et al., 2017*; *Rinne et al., 2015*). Briefly, sections were incubated in 10 mM sodium citrate buffer (pH 6) for 20 min in a pressure cooker for antigen retrieval. Thereafter, sections were quenched in 1% $H_2O_2$ for 10 min and blocked in 5% normal horse serum containing 1% BSA. After blocking, sections were incubated overnight with a primary antibody against MC1-R (Elabscience, Texas, USA, # E-AB-15765) followed by biotinylated horseradish peroxidase-conjugated secondary antibody incubation and detection with diaminobenzidine (ABC kit, Vector Labs, Burlingame, USA). For isotype control, a consecutive heart section was treated similarly except that the primary MC1-R antibody was replaced by purified normal rabbit IgG (Novus Biologicals, Littleton, CO, USA, # NB810-56910). For immunofluorescence, liver sections were incubated overnight with antibodies against MC1-R (Elabscience) and serum albumin (Bioss Antibodies, # BSM-0945M), cytokeratin 19 (CK-19, Novus Biologicals, # NBP2-44827), CD31 (R&D Systems, # AF3628), or Mac-2 (Cedarlane labs, # CL8942AP) followed by detection with fluorochrome-conjugated secondary antibodies (anti-rabbit Alexa Fluor 647 and anti-mouse, anti-rat or anti-goat Alexa Fluor 488, Invitrogen). To visualize hepatic lipid content, a transverse piece of the left liver lobe was embedded in O.C.T. compound (Tissue-Tek), cryosectioned and stained with Oil Red O. Sections were counterstained with hematoxylin (CarlRoth) or DAPI (Fluoroshield mounting medium, abcam), cover-slipped and then scanned with Pannoramic 250 or Pannoramic Midi digital slide scanner (3DHISTECH Kft, Budapest, Hungary).

## RNA isolation, cDNA synthesis, and quantitative RT-PCR

HepG2 cell samples were collected into QIAzol Lysis Reagent and total RNA was extracted using Direct-zol RNA Miniprep (Zymo Research, CA, USA). Liver samples were first homogenized in QIAzol Lysis Reagent (Qiagen) using the Qiagen TissueLyser LT Bead Mill (QIAGEN, Venlo, Netherlands). Total RNA from each sample was extracted and reverse-transcribed to cDNA with PrimeScript RT reagent kit (Takara Clontech) according to the manufacturer's instructions. The RNA quality and concentration were evaluated by Nanodrop. Quantitative real-time polymerase chain reaction (RT-PCR) was

performed using SYBR Green protocols (Kapa Biosystems, MA, USA) on a real-time PCR detection system (Applied Biosystems 7300 Real-Time PCR system). Each sample was run in duplicate. Target gene expression was normalized to the geometric mean of two housekeeping genes (β-actin and ribosomal protein S29 or GAPDH) using the delta-Ct method and results are presented as relative transcript levels ($2^{-\Delta\Delta Ct}$). Primer sequences are presented in *Tables 1 and 2*.

## Immunoblotting

Liver and HepG2 samples were lysed in RIPA buffer (50 mM NaCl, 1% Triton X-100, 0.5% Sodium deoxycholate, 0.1% SDS, pH 8.0) supplemented with protease and phosphatase inhibitor cocktail (ThermoFisher, #A32961). Liver samples were additionally homogenized using the Qiagen TissueLyser LT Bead. Equal amounts (30 µg) of total protein were separated by 10% SDS-polyacrylamide gel electrophoresis (SDS-PAGE) and transferred onto nitrocellulose membranes (GE Healthcare). After blocking with Tris-Buffered Saline (Sigma-Aldrich) containing 0.1% Tween 20 detergent (Sigma-Aldrich) and 5% skimmed milk (Carl Roth) for 1 hr at room temperature (RT), membranes were incubated with specific primary antibodies for MC1-R (Alomone Labs, #AMR-025), LDLR (Novus Biologicals, Littleton, CO, USA, #NBP1-06709), SR-BI (NovusBio, #NB400-104), SREBP2 (Novus Biologicals, #NB100-74543), HMGCR (Novus Biologicals, #NBP2-66888), DHCR7 (abcam, #ab103296), MRP4 (Cell Signaling Tech, Frankfurt, DE, #12857), StAR (Cell Signaling Tech, #8449), CYP8B1 (St John's Laboratory Ltd, #STJ92607), phospho-AMPKα (Cell Signaling Tech, #2535), and AMPKα (Cell Signaling Tech, #2532) over-night at +4 °C. Next day, membranes were washed and incubated with Horseradish peroxidase (HRP)-conjugated secondary antibodies (Cell Signaling Tech) for 1 hr at RT. Proteins were visualized using a chemiluminescence (ECL) kit (Millipore, MA, USA). Target protein expression was normalized to β-actin (Sigma-Aldrich, #2066) or vinculin (Bio-Rad, #MCA465GA) to correct for loading, and band densities were analyzed using ImageJ software (NIH, Bethesda, MD, USA).

## Plasma and liver extract analyses

Plasma samples were obtained from EDTA-anticoagulated whole blood after centrifugation. Plasma total cholesterol and triglyceride concentrations were determined using enzymatic colorimetric assays (CHOD-PAP and GPO-PAP, mtiDiagnostics, Idstein, Germany) according to the manufacturer's protocols. For the determination of hepatic lipid content, liver samples (~100 mg) were homogenized in 500 µl of PBS with 0.1% NP-40 (Abcam) using TissueLyser and then centrifuged to remove insoluble components (*Nuutinen et al., 2018*; *Rinne et al., 2018*). Cholesterol and triglycerides concentrations were quantified in the liver homogenates using CHOD-PAP and GPO-PAP reagents.

## Bile acid measurements

Bile acids (BA) were measured in plasma, liver, and fecal samples. BAs were extracted and analyzed by an ultra-high-performance liquid chromatography tandem-mass spectrometry method (UHPLC-MS/MS) as previously described (*Jäntti et al., 2014*). The order of the samples was randomized before sample preparation. The BAs analysed were Litocholic acid (LCA), 12-oxo-litocholic acid (12-oxo-LCA), Chenodeoxycholic acid (CDCA), Deoxycholic acid (DCA), Hyodeoxycholic acid (HDCA), Ursodeoxycholic acid (UDCA), Dihydroxycholestanoic acid (DHCA), 7-oxo-deoxycholic acid (7-oxo-DCA), 7-oxo-hyocholic acid (7-oxo-HCA), Hyocholic acid(HCA), β-Muricholic acid (β-MCA), Cholic acid (CA), $\omega$/α-Muricholic acid ($\omega$/α-MCA), Glycolitocholic acid (GLCA), Glycochenodeoxycholic acid (GCDCA), Glycodeoxycholic acid (GDCA), Glycohyodeoxycholic acid (GHDCA), Glycoursodeoxycholic acid (GUDCA), Glycodehydrocholic acid (GDHCA), Glycocholic acid (GCA), Glycohyocholic acid (GHCA), Taurolitocholic acid (TLCA), Taurochenodeoxycholic acid (TCDCA), Taurodeoxycholic acid (TDCA), Taurohyodeoxycholic acid (THDCA), Tauroursodeoxycholic acid (TUDCA), Taurodehydrocholic acid (TDHCA), Tauro-α-muricholic acid (TαMCA), Tauro-β-muricholic acid (TβMCA), Taurocholic acid (TCA), Trihydroxycholestanoic acid (THCA), and Tauro-$\omega$-muricholic acid (T$\omega$MCA).

For plasma sample analysis, BAs were extracted by adding 20 µl plasma to 200 µl crash solvent methanol containing 25 ppb LCA-d4 and 2.5 ppb each of the internal standards (TCA-d4, GUDCA-d4, GCA-d4, CA-d4, UDCA-d4, GCDCA-d4, CDCA-d4, DCA-d4, and GLCA-d4) and filtering them using a Supelco protein precipitation filter plate. A 170 uL aliquot of the supernatant was dried under a gentle flow of nitrogen and resuspended using 20 µL resuspension solution (Methanol:water (40:60) with 5 ppb Perfluoro-n-[13C9]nonanoic acid as in injection standard). Blank samples were prepared

**Table 1.** Quantitative RT-PCR primers for mouse genes.

| Gene name<br>Accession number | 5'–3' primer sequence |
|---|---|
| *Actb*<br>NM_007393.5 | Forward: tccatcatgaagtgtgacgt<br>Reverse: gagcaatgatcttgatcttca |
| *Akr1d1*<br>NM_145364.2 | Forward: gaaaagatagcagaagggaaggt<br>Reverse: gggacatgctctgtattccataa |
| *Bsep*<br>NM_021022.3 | Forward: aagctacatctgccttagacacagaa<br>Reverse: caatacaggtccgaccctctct |
| *Ccl2*<br>NM_011333.3 | Forward: aggtccctgtcatgcttctg<br>Reverse: aaggcatcacagtccgagtc |
| *Col1a1*<br>NM_007742.4 | Forward: gctcctcttaggggccact<br>Reverse: ccacgtctcaccattgggg |
| *Cyp7a1*<br>NM_007824.2 | Forward: gatcctctgggcatctcaag<br>Reverse: agaggctgctttcattgctt |
| *Cyp7b1*<br>NM_007825.4 | Forward: gaaaactcttcaaaggcaacatgg<br>Reverse: actggaaagggttcagaacaaatg |
| *Cyp8b1*<br>NM_010012.3 | Forward: gccttcaagtatgatcggttcct<br>Reverse: gatcttcttgcccgacttgtaga |
| *Cyp27a1*<br>NM_024264.5 | Forward: gcctcacctatgggatcttca<br>Reverse: tcaaagcctgacgcagatg |
| *Fxr*<br>NM_001163700.1 | Forward: tccggacattcaaccatcac<br>Reverse: tcactgcacatcccagatctc |
| *Hnf4a*<br>NM_008261.3 | Forward: accaagaggtccatggtgttt<br>Reverse: gtgccgagggacgatgtag |
| *Hsd3b7*<br>NM_133943.2 | Forward: gggagctgcgtgtctttga<br>Reverse: gtggatggtctttggactggc |
| *Il1b*<br>NM_008361.4 | Forward: tgtaatgaaagacggcacacc<br>Reverse: tcttctttgggtattgcttgg |
| *Il6*<br>NM_031168.2 | Forward: ggccttccctacttcacaag<br>Reverse: atttccacgatttcccagag |
| *Lrh1*<br>NM_030676.3 | Forward: tgggaaggaagggacaatctt<br>Reverse: cgagactcaggaggttgttgaa |
| *Abcc3 (Mrp3)*<br>NM_029600.4 | Forward: ctgggtccctgcatctac<br>Reverse: gccgtcttgagcctggataac |
| *Abcc4 (Mrp4)*<br>NM_001163676.1 | Forward: ggcactccggttaagtaactc<br>Reverse: tgtcacttggtcgaatttgttca |
| *Ntcp*<br>NM_011387.2 | Forward: gaagtccaaaaggccacactatgt<br>Reverse: acagccacagagagggagaaag |
| Gene name<br>Accession number | 5'–3' primer sequence |
| *Slc51a (Osta)*<br>NM_145932.3 | Forward: aggcaggactcatatcaaacttg<br>Reverse: tgagggctatgtccactggg |
| *S29*<br>NM_009093.2 | Forward: atgggtcaccagcagctcta<br>Reverse: agcctatgtccttcgcgtact |
| *Nr0b2(Shp)*<br>NM_011850.3 | Forward: tgggtcccaaggagtatgc<br>Reverse: gctccaagacttcacacagtg |
| *Stard1*<br>NM_011485.5 | Forward: atgttcctcgctacgttcaag<br>Reverse: cccagtgctctccagttgag |
| *Tgfb1*<br>NM_011577.2 | Forward: ccgcaacaacgccatctatg<br>Reverse: cccgaatgtctgacgtattgaag |
| *Tnf*<br>NM_013693.3 | Forward: ctgaacttcggggtgatcgg<br>Reverse: ggcttgtcactcgaattttgaga |

**Table 2.** Quantitative RT-PCR primers for human genes.

| Gene name<br>Accession number | 5'–3' primer sequence |
| --- | --- |
| *ACTA1*<br>NM_001100.4 | Forward: aggtcatcaccatcggcaacga<br>Reverse: gctgttgtaggtggtctcgtga |
| *ACTA2*<br>NM_001613.4 | Forward: ctatgcctctggacgcacaact<br>Reverse: cagatccagacgcatgatggca |
| *ACTB*<br>NM_001101.5 | Forward: caccattggcaatgagcggttc<br>Reverse: aggtctttgcggatgtccacgt |
| *COL1A1*<br>NM_000088.4 | Forward: gagggccaagacgaagacatc<br>Reverse: cagatcacgtcatcgcacaac |
| *GAPDH*<br>NM_002046.7 | Forward: tcaaggctgagaacgggaag<br>Reverse: cgccccacttgattttggag |
| *IL6*<br>NM_00600.5 | Forward: gatgagtacaaaagtcctgatcca<br>Reverse: ctgcagccactggttctgt |
| *LDLR*<br>NM_000527.5 | Forward: ccacggtggagatagtgaca<br>Reverse: ctcacgctactgggcttctt |
| *SCARB1*<br>NM_005505.5 | Forward: ctggcagaagcggtgact<br>Reverse: cagagcagttcatgggggatt |
| *TGFB1*<br>NM_000660.7 | Forward: tacctgaacccgtgttgctctc<br>Reverse: gttgctgaggtatcgccaggaa |
| *TNF*<br>NM_000594.4 | Forward: cctctctctaatcagccctctg<br>Reverse: gaggacctgggagtagatgag |

by pipetting 200 µl crash solvent into a vial, then drying and resuspending them the same way as the other samples. Calibration curves were prepared by pipetting 20 µl of standard dilution into vials, adding 200 µL crash solution, and drying and resuspending them in the same way as the other samples. The concentrations of the standard dilutions were between 0.0025 and 600 ppb.

Aliquots of 25 mg of liver were powdered using dry ice under a glass plate and transferred to a plastic tube. Water (100 µl) was added and the sample was sonicated for 30 s. For protein precipitation and extraction, 300 ul of acetonitrile was added and the samples were filtered using a Supelco protein precipitation filter plate. A 255 ul aliquot of the supernatant was dried under a gentle flow of nitrogen and resuspended using a 20 µl resuspension solution.

Feces were collected over 48 hr from individually housed mice. All fecal samples were freeze-dried prior to extraction to determine the dry weight and 20 µl of water was added for each mg of dry weight in the sample. The samples were homogenized by adding homogenizer beads, freezing samples to at least –70 °C, and homogenizing them for 5 min using a bead beater. BAs were extracted by adding 40 µl fecal homogenate to 400 µl crash solvent (methanol containing 62, 5 ppb each of the internal standards) and filtering them using a Supelco protein precipitation filter plate. The samples were dried and resuspended the same way as serum samples. QCs, blanks, and calibration curves were prepared similarly to plasma samples.

The LC separation was performed on a Sciex Exion AD 30 (AB Sciex Inc, Framingham, MA) LC system consisting of a binary pump, an autosampler set to 15 °C, and a column oven set to 35 °C. A waters Aquity UPLC HSS T3 (1.8 µm, 2.1 × 100 mm) column with a precolumn with the same material. Eluent A was 0.1% formic acid in water and eluent B was 0.1% formic acid in methanol. The gradient started from 15% B and increased to 30% B over 1 min. The gradient further increased to 70% B over 15 min. The gradient was further increased to 100% over 2 min. The gradient was held at 100% B for 4 min then decreased to 15% B over 0.1 min and re-equilibrated for 7.5 min. The flow rate was 0.5 mL/min and the injection volume was 5 µL.

The mass spectrometer used for this method was a Sciex 5500 QTrap mass spectrometer operating in scheduled multiple reaction monitoring mode in negative mode. The ion source gas 1 and 2 were both 40 psi. The curtain gas was 25 psi, the CAD gas was 12 and the temperature was 650 °C. The spray voltage was 4500 V. Data processing was performed on Sciex MultiQuant.

## Filipin staining of cellular free cholesterol

HepG2 cells were seeded (40,000 cells/well) on 96-well plates (PhenoPlate, PerkinElmer) and grown until the cells reached 70% confluency. Thereafter, cells were first serum-deprived (0.5% FBS) for 16 hr, and then treated with either α-MSH or LD211 for 24 hr. After the treatment, cells were washed with PBS and fixed with 4% paraformaldehyde (Sigma-Aldrich) for 15 min at room temperature. Cells were subsequently washed with PBS and incubated with 1.5 mg/ml glycine (Sigma-Aldrich, 10 min at RT) to quench unreacted paraformaldehyde followed by washing with PBS. Cells were stained with 50 ug/ml of Filipin (Sigma-Aldrich, #F9765) for 1 hr at 37 °C and washed with PBS. Fluorescence signal was measured with EnSight Multimode Plate Reader (PerkinElmer) with the 360 nm excitation and 480 emission wavelengths.

## LDL and HDL uptake assay

HepG2 cells were seeded (40,000 cells/well) on 96-well plates (PhenoPlate, PerkinElmer), serum-deprived (0.5% FBS) for 16 hr and then treated with α-MSH or LD211 for 18 hr at 37 °C. After the treatment, cells were washed with PBS and incubated with fluorescently-labeled HDL (Dil-HDL, 20 µg/ml, CliniSciences) or LDL (Dil-LDL, 10 µg/ml, CliniSciences) for 4 hr at 37 °C. After the incubation, cells were again washed with PBS and the fluorescence signal was measured with EnSight Multimode Plate Reader (PerkinElmer) with the 549 nm excitation and 565 emission wavelengths and normalized against cell confluency.

## Flow cytometric analysis of cell surface LDLR expression

HepG2 cells were treated as indicated in the figure legends, washed with PBS, and detached using EDTA. To quantify the expression of LDLR on the cell surface, HepG2 cells were stained with PE-conjugated anti-human LDLR antibody (clone C7, BD Biosciences) and then analyzed with LSR Fortessa (BD Biosciences) and FlowJo software (FlowJo, LLC, Ashland, USA).

## Cyclic AMP determination

To measure intracellular cAMP concentrations, HepG2 cells were pretreated with 3-isobutyl-1-methylxanthine (0.1 mM, IBMX, Sigma-Aldrich) for 30 min and then stimulated with α-MSH or the selective MC1-R agonist LD211 (0.1 nM, 10 nM or 1 µM) for 30 min. Cells were thereafter lysed with 0.1 M HCl and assayed for cAMP levels with a commercial kit (Cyclic AMP Select ELISA kit, Cayman Chemical, #501040) according to the manufacturer's instructions. Results were normalized against total protein concentrations (Pierce BCA Protein Assay Kit, Thermo Fisher) and expressed as a percentage of control samples that were left untreated.

## Enzyme-linked immunosorbent assays (ELISA) of phosphorylated ERK1/2 and JNK

HepG2 cells were stimulated with α-MSH or the selective MC1-R agonist LD211 as indicated in the figure legends. Cells were thereafter lysed with Lysis Buffer #6 (R&D Systems) and assayed for the expression levels of phospho-ERK1 (T202/Y204)/ERK2 (T185/Y187) and phospho-JNK (T183/Y185 for JNK1/2 and T221/Y223 for JNK3) with commercial kits (DuoSet IC ELISA, R&D Systems, # DYC1018B and # DYC1387B) according to manufacturer's instructions. Results were normalized against total protein concentrations (Pierce BCA Protein Assay Kit, ThermoFisher).

## Statistics

All experiments were performed at least three independent times. Statistical analyses were performed with GraphPad Prism 9 software (La Jolla, CA, USA). Statistical significance between the experimental groups was determined by two-tailed, unpaired Student's t-test or one-way or two-way ANOVA followed by Dunnet *post hoc* tests. The D'Agostino and Pearson omnibus normality test method was utilized to check the normality of the data. Possible outliers in the data sets were identified using the regression and outlier removal (ROUT) method of Q-level of 1%. Data are expressed as mean ± standard error of the mean (SEM). Results were considered significant for $p < 0.05$.

## Acknowledgements

We thank Sanna Bastman, Salla Juhantalo, and Johanna Jukkala for their excellent technical support. The histological methods were performed by the Histology core facility of the Institute of Biomedicine, University of Turku, Finland. BA measurements were performed at the Turku Metabolomics Centre (Turku Bioscience Centre, Turku, Finland) with the support of Biocenter Finland. Turku Center for Disease Modeling, TCDM, Turku, Finland, (https://www.tcdm.fi/), a member of Biocenter Finland, is acknowledged for setting up the genotyping protocol for Mc1r LKO mice. This work was supported by the Academy of Finland (grant 315351 to PR); the Sigrid Jusélius Foundation (to PR); the Finnish Cultural Foundation (to PR), the Finnish Foundation for Cardiovascular Research (to ES and PR); and the National Institutes of Health (grant GM-104080 to MC).

## Additional information

### Funding

| Funder | Grant reference number | Author |
|---|---|---|
| Academy of Finland | 315351 | Petteri Rinne |
| Sigrid Juséliuksen Säätiö | | Petteri Rinne |
| Suomen Kulttuurirahasto | | Petteri Rinne |
| The Finnish Foundation for Cardiovascular Research | | Eriika Savontaus Petteri Rinne |
| National Institutes of Health | GM-104080 | Minying Cai |

The funders had no role in study design, data collection and interpretation, or the decision to submit the work for publication.

### Author contributions

Keshav Thapa, Conceptualization, Data curation, Formal analysis, Investigation, Visualization, Methodology, Writing - original draft; James J Kadiri, Karla Saukkonen, Iida Pennanen, Investigation; Bishwa Ghimire, Formal analysis, Methodology; Minying Cai, Resources, Funding acquisition, Writing - review and editing; Eriika Savontaus, Conceptualization, Supervision, Funding acquisition, Writing - review and editing; Petteri Rinne, Conceptualization, Data curation, Formal analysis, Supervision, Funding acquisition, Investigation, Visualization, Methodology, Writing - original draft, Writing - review and editing

### Author ORCIDs

Karla Saukkonen http://orcid.org/0000-0002-8767-2074
Petteri Rinne http://orcid.org/0000-0001-6916-8627

### Ethics

The animal experiments were approved by the local ethics committee (Animal Experiment Board in Finland, License Numbers: ESAVI/6280/04.10.07.2016 and ESAVI/1260/2020) and conducted in accordance with the institutional and national guidelines for the care and use of laboratory animals.

### Decision letter and Author response

Decision letter https://doi.org/10.7554/eLife.84782.sa1
Author response https://doi.org/10.7554/eLife.84782.sa2

## Additional files

### Supplementary files
• MDAR checklist

### Data availability

All data generated or analysed during this study are included in the manuscript and supporting file. Source Data files have been provided for Main Figures and Supplementary Figures.

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
