## [Editor Report]

The significance of this manuscript is that it provides useful information for the field of hepatology and endocrinology on the regulatory mechanisms of cholesterol homeostasis by melanocortin. The authors provide solid evidence utilizing both in vivo and in vitro molecular, cellular, and biochemical approaches to support their claims.

---

## [Decision Letter]

**Decision letter after peer review:**

Thank you for submitting your article "Melanocortin 1 receptor regulates cholesterol and bile acid metabolism in the liver" for consideration by *eLife*. Your article has been reviewed by 2 peer reviewers, one of whom is a member of our Board of Reviewing Editors, and the evaluation has been overseen by Carlos Isales as the Senior Editor. The following individual involved in the review of your submission has agreed to reveal their identity: Julia Sánchez-Ceinos (Reviewer #2).

Essential revisions:

1) Justify the use of only female mice for in vivo studies.

2) Colocalization studies by confocal microscopy of MC1-R and different cell type markers (hepatocytes, macrophages, endothelial cells, fibroblasts, cholangiocytes.

3) Discuss the potential roles of MC1R in cholangiocytes in the modulation of bile acid homeostasis.

4) Validate hepatic MC1R knockdown at the protein level.

5) Determine the effects of HFD on disease progression in hepatic MC1R KO mice, or, determine the therapeutic potential of LD211 or MSH in reversing diet-induced obesity.

*Reviewer #1 (Recommendations for the authors):*

1) Assess MC1R expression in cholangiocytes via IHC.

2) Discuss the potential roles of MC1R in cholangiocytes in the modulation of bile acid homeostasis.

3) Validate hepatic MC1R knockdown at the protein level.

4) Protein validation for mRNA changes in hepatic genes involved in cholesterol and fatty acid synthesis.

5) Assess the cholesterol-lowering effect of LD211 in the presence of AMPK inhibitors.

6) Determine the effects of HFD on disease progression in hepatic MC1R KO mice.

7) Determine the therapeutic potential of LD211 or MSH in reversing diet-induced obesity.

*Reviewer #2 (Recommendations for the authors):*

Below follows specific questions and comments:

1) Please justify the use of only female mice for in vivo studies.

2) Although immunohistochemistry images in Figure 1A demonstrating the presence of MC1-R in the liver are convincing and understanding that since hepatocytes represent the more abundant cell type in this organ is plausible to conclude that MC1-R is mainly expressed in these cells, these images do not fully support this conclusion. As mentioned in the introduction (lines 84-87), MC1-R has been also demonstrated to be expressed in a variety of peripheral cells. Therefore, to unequivocally demonstrate that MC1-R is mainly expressed in these cells colocalization studies by confocal microscopy of MC1-R and different cell type markers (hepatocytes, macrophages, endothelial cells, fibroblasts…) are required. Without a robust expression of MC1-R exclusively in hepatocytes, further mechanistic studies are not fully justified.

3) Concerns regarding the specificity of the antibody used for both immunohistochemistry images in Figure 1A and western blotting analysis in Figure 1C and Supplementary Figure 1 might rise. As a receptor, MC1-R expected localization would be in the plasma membrane whereas on the presented histological images appears to have uniform immunostaining all over the cytoplasm. Likewise, blots in Supplementary Figure 1 show diverse unspecific bands. In this figure, is the lack of bands in HepG2 protein extract due to the species-specificity of the MC1-R antibody? What is the reason behind the use of mouse heart samples in this study?

4) If experimentally possible and/or publicly available data, expression levels of MC1-R in human liver samples from healthy vs. liver disease (hypercholesterolemia, NASH, NAFLD…) will further strengthen your findings and add valuable translational value to your work.

5) The observation of that liver from L-MC1-R-/- mice exhibit signs indicative of fibrosis, but no changes were observed in pro-inflammatory genes is intriguing and deserves additional investigation. Does the expression of the fibrosis- and inflammation-related genes tested in the mice changes in the HepG2 under the different treatments employed?

6) Liquid chromatography-mass spectrometry data from Figure 3 and Supplementary Figures 4 and 5A should be provided as a supplementary table and raw data deposited in an appropriate public data repository with the corresponding mention in the method section. Moreover, a more detailed description of this methodology would be highly appreciated by the readers.

7) What is the biological relevance of the ratio of CA:CDCA observed in Figure 3G and Supplementary Figure 5B?

8) The dysregulation of genes involved in bile acid synthesis, uptake/secretion of bile acids, and nuclear receptors in the liver from KO mice is convincing at the mRNA level in Figures 4A-C. At the protein level (Figures 4D and E) it is unclear. Only two of these proteins were confirmed by western blotting studies (and another protein showed contradictory results). These studies are clearly insufficient in order to draw any valid conclusion. The potential changes of a meaningful number of genes should be validated by alternative methods.

9) Blot images presented in Figures 4D, 5A-C, and 5J, and Supplementary Figures 6C-D and 7E do not seem to have quality enough and/or be representative of their corresponding quantifications. Please replace them in accordance.

10) The use of HepG2 cells for the in vitro studies is not enough justified. This cellular model is not ideal for the study of metabolic disturbances and does not match some of the criteria used in the in vivo model (they are derived from a male child with hepatocellular carcinoma). It may be understood that they were used for initial screening purposes but selected, key aspects, of the study should be checked in primary hepatocytes from L-MC1-R-/- and control mice.

11) The decrease in MC1-R in HepG2 in response to palmitic acid, but not when exposed to LDL or atorvastatin, is an interesting observation. Based on what did you choose the selected concentration of these compounds to mimic the cholesterol-rich diet administrated to the animals?

12) The authors also show that α-MSH treatment reduced the lipid metabolism in hepatocytes. However, it is unclear whether the pathological mechanisms mediated by the downregulation of MC1-R in hepatocytes would be restored by the addition of α-MSH. Additional studies in which MC1-R expression can be manipulated may be needed. Similarly, it would also be worthwhile to use clinically approved melanocortin receptor-targeted drugs in these experiments.

13) The attempt to elucidate the intracellular signalling cascades that might be behind the beneficial effects of MC1-R in the liver is mostly appreciated. Nevertheless, in the present form, these investigations fail to pinpoint a specific mechanism leading to inconclusive results. It would be more convincing if the authors can perform transcriptomics or proteomics in primary hepatocytes from L-MC1-R-/- and control animals to pursue potential targets.

---

## [Author Response]

Essential revisions:1) Justify the use of only female mice for in vivo studies.

The use of female mice was based on two main reasons. First, body composition was measured by quantitative NMR scanning, which according to our own experience, can provoke aggressive behavior in group-housed male mice due to cross-contamination of odors (etc. urine) between cages. Second, we had previously observed that pharmacological activation of MC1-R reduced plasma cholesterol level and induced hepatic *Ldlr* expression in female *Apoe^-/-^* mice (as discussed on manuscript page 12, lines 475-478), which further supported the preferential use of female mice in the present study.

2) Colocalization studies by confocal microscopy of MC1-R and different cell type markers (hepatocytes, macrophages, endothelial cells, fibroblasts, cholangiocytes.

To obtain a preliminary view on the localization of MC1-R in the liver, we used a publicly available resource (www.livercellatlas.org) to screen for *MC1R* expression in liver single cell RNA-sequencing dataset. Among sequenced nonleukocyte (CD45^-^) cells, *MC1R*-positive cell were mainly hepatocytes but also cholangiocytes, hepatic stellate cells and endothelial cells (Author response image 1 upper panel). In terms of distribution in myeloid cells, *MC1R*-positive cells were more scarce compared to distribution in CD45^-^ cell populations but were spread widely in different monocyte and Kupffer cell (KC) populations (Author response image 1 lower panel).

**Author response image 1. sa2fig1:** 

In general, the number of *MC1R*-positve cells was relatively low and probably underestimated due to the low detection sensitivity of scRNA-seq technology and consequent dropout of low-expression genes such as *MC1R*.Nevertheless, the mining of this scRNA-seq dataset gave a good starting point for studying the localization of MC1-R in the mouse liver by double immunofluorescence staining. In line with the expression pattern of *MC1R* in the human liver, the colocalization studies demonstrate the presence of MC1-R in albumin-expressing hepatocytes (Figure 1B). In addition, MC1-R expression colocalized in CK19positive cholangiocytes and Mac2-positive monocytes/macrophages in the mouse liver, but not substantially in CD31-positive endothelial cells (Figure 1—figure supplement 1). These findings open the possibility for investigating the role of MC1R in other cell types of the liver in future studies. Importantly, supporting our initial hypothesis that MC1-R is involved in the regulation of cholesterol and bile acid metabolism in the liver, MC1-R was found to mainly colocalize in hepatocytes and cholangiocytes that are major regulators of cholesterol homeostasis and bile acid synthesis.

3) Discuss the potential roles of MC1R in cholangiocytes in the modulation of bile acid homeostasis.

This is indeed an important aspect of the study, which is supported by the finding of MC1-R expression in cholangiocytes (Figure 1—figure supplement 1). We have now discussed the potential role of MC1-R in this regard (page 12, lines 456462).

4) Validate hepatic MC1R knockdown at the protein level.

As requested, we quantified MC1-R protein expression by Western blotting in the liver of L-Mc1r^-/-^ mice. MC1-R protein level was significantly reduced in L-Mc1r^-/-^ mice compared to L-Mc1^+/-^ mice that were used as controls (Figure 1 figure supplement 3). However, there was considerable residual MC1-R expression in L-Mc1r^-/-^ mice, which most likely reflects the presence of MC1-R in nonhepatocytes as discussed above (response to comment #2).

5) Determine the effects of HFD on disease progression in hepatic MC1R KO mice, or, determine the therapeutic potential of LD211 or MSH in reversing diet-induced obesity.

As suggested, we phenotyped the hepatic MC1R KO (L-Mc1r^-/-^) mice after feeding them a cholesterol- and fat-rich Western diet for 12 weeks (RD Western Diet, D12079B, Research Diets Inc, NJ, USA). This was exactly the same dietary regimen (product and duration) that was used to study the changes in hepatic MC1-R expression in wild-type C57Bl mice (Figure 1C and D). We observed that 12-week Western diet feeding induced a significant gain in body weight and total fat mass as well as an increase in plasma and hepatic cholesterol and TG levels (Figure 2—figure supplement 2). L-Mc1r^-/-^ mice did not show a difference in body weight gain, but the weight gain was attributable to enhanced gain in fat mass and a blunted increase in lean mass compared to control Mc1r^fl/fl^ mice (Figure 2—figure supplement 2A, D and E). Furthermore, liver weight and plasma cholesterol and TG concentrations were unchanged in HFD-fed L-Mc1r^-/-^ mice (Figure 2—figure supplement 2B, C, F and G). Importantly, recapitulating the phenotype observed in chow-fed mice, hepatic cholesterol and TG content was significantly increased in L-Mc1r^-/-^ mice after a HFD challenge (Figure 2—figure supplement 2H and I). Taken together, it appears that the phenotype of HFD-fed L-Mc1r^-/-^ mice was slightly diluted compared to the phenotype observed in chow-fed L-Mc1r^-/-^ mice. This phenotypic difference might relate to the finding that Western diet feeding reduced the hepatic expression of MC1-R, thus limiting the incremental effect of genetically induced MC1-R deficiency on hypercholesterolemia and hepatic lipid accumulation.

We have previously studied the effects of pharmacological MC1-R activation in

Western diet-fed *Apoe^-/-^* mice and observed that chronic treatment with a selective MC1-R agonist reduced plasma cholesterol level and upregulated hepatic *Ldlr* expression without affecting body weight gain (*Rinne P et al., Circulation. 2017 Jul 4;136(1):83-97.*). These findings are also discussed on manuscript page 12/ lines 475478. Although the selective MC1-R agonist was different in that particular study, it is expected that LD211 would also elicit a similar cholesterol-lowering effect in Western diet-fed mice. Chronic treatment with a-MSH, on the other hand, would likely produce wide-ranging metabolic effects. In addition to MC1-R activation in hepatocytes and its consequent effect on liver cholesterol metabolism, a-MSH would affect feeding, energy expenditure and cholesterol metabolism *via* MC4-R activation in the central nervous system as well as fatty acid and glucose metabolism *via* MC5-R activation in the skeletal muscle. Therefore, the phenotype associated with a-MSH treatment would be complex and mediated by multiple mechanisms and MC-R subtypes, thus making it difficult to interpret the exact contribution of hepatic MC1-R signaling to the observed phenotype.

Reviewer #1 (Recommendations for the authors):1) Assess MC1R expression in cholangiocytes via IHC.

As requested, we did a double immunofluorescence staining using MC1-R antibody in combination with an antibody against cytokeratin 19 (CK-19) (*He L et al., Nat. Med. 2017, 23, 1488–1498 and Sekiya S et al., Am J Pathol. 2014 May; 184(5):1468-78*). Figure 1—figure supplement 1 demonstrates the presence of MC1R also in CK19-expressing cholangiocytes.

2) Discuss the potential roles of MC1R in cholangiocytes in the modulation of bile acid homeostasis.

As suggested by the reviewer, we have now discussed the potential role of MC1-R in cholangiocytes (page 12, lines 456-462).

3) Validate hepatic MC1R knockdown at the protein level.

As requested, we quantified MC1-R protein expression by Western blotting in the liver of L-Mc1r^-/-^ mice. MC1-R protein level was significantly reduced in L-Mc1r^-/-^ mice compared to L-Mc1^+/-^ mice that were used as controls (Figure 1figure supplement 3).

4) Protein validation for mRNA changes in hepatic genes involved in cholesterol and fatty acid synthesis.

We quantified the protein expression of SREBP2, HMGCR and DHCR 7 in the liver samples by Western blotting. The results are presented in Figure 2F-H and discussed in more detail above (response to comment 2).

5) Assess the cholesterol-lowering effect of LD211 in the presence of AMPK inhibitors.

As requested, we determined the effect of LD211 on cellular cholesterol level in the presence of the AMPK inhibitor dorsomorphin. Consistently, LD211 significantly reduced cholesterol level in HepG2 cells, while pretreatment of the cells with dorsomorphin completely abolished the cholesterol-lowering effect of LD211 (Figure 7—figure supplement 2).

6) Determine the effects of HFD on disease progression in hepatic MC1R KO mice.

As requested, we fed MC1R KO (L-Mc1r^-/-^) and control a cholesterol- and fat-rich Western diet for 12 weeks (RD Western Diet, D12079B, Research Diets Inc, NJ, USA) and phenotyped them for body weight gain and composition as well as liver and plasma lipid profiles. This was exactly the same dietary regimen (product and duration) that was used to study the changes in hepatic MC1-R expression in wildtype C57Bl mice (Figure 1C and D). We observed that 12-week Western diet feeding induced a significant gain in body weight and total fat mass as well as an increase in plasma and hepatic cholesterol and TG levels (Figure 2—figure supplement 2). LMc1r^-/-^ mice did not show a difference in body weight gain, but the weight gain was attributable to enhanced gain in fat mass and a blunted increase in lean mass compared to control mice (Figure 2—figure supplement 2A, D and E). Furthermore, liver weight and plasma cholesterol and TG concentrations were unchanged in HFDfed L-Mc1r^-/-^ mice (Figure 2—figure supplement 2B, C, F and G). Importantly, recapitulating the phenotype observed in chow-fed mice, hepatic cholesterol and TG content was significantly increased in L-Mc1r^-/-^ mice after a HFD challenge (Figure 2—figure supplement 2H and I). Taken together, it appears that the phenotype of HFDfed L-Mc1r^-/-^ mice was slightly diluted compared to the phenotype observed in chowfed L-Mc1r^-/-^ mice. This phenotypic difference might relate to the finding that Western diet feeding reduced the hepatic expression of MC1-R, thus limiting the incremental effect of genetically induced MC1-R deficiency on hypercholesterolemia and hepatic lipid accumulation.

7) Determine the therapeutic potential of LD211 or MSH in reversing diet-induced obesity.

We have previously studied the effects of pharmacological MC1-R activation in Western diet-fed mice and observed that chronic treatment with a selective MC1-R agonist reduced plasma cholesterol level and upregulated hepatic *Ldlr* expression without affecting body weight gain (*Rinne P et al., Circulation. 2017 Jul 4;136(1):83-97.*). These findings are also discussed on manuscript page 12/lines 475-479. Although the selective MC1-R agonist was different in that particular study, it is expected that LD211 would also elicit a similar cholesterol-lowering effect in Western diet-fed mice. Chronic treatment with a-MSH, on the other hand, would likely produce wide-ranging metabolic effects. In addition to MC1-R activation in hepatocytes and its consequent effect on liver cholesterol metabolism, a-MSH would affect feeding, energy expenditure and cholesterol metabolism *via* MC4-R activation in the central nervous system as well as fatty acid and glucose metabolism *via* MC5-R activation in the skeletal muscle.

Therefore, the phenotype associated with a-MSH treatment would be complex and mediated by multiple mechanisms and MC-R subtypes, thus making it difficult to interpret the exact contribution of hepatic MC1-R signaling to the observed phenotype.

Reviewer #2 (Recommendations for the authors):Below follows specific questions and comments:1) Please justify the use of only female mice for in vivo studies.

The use of female mice was based on two main reasons. First, body composition was measured by quantitative NMR scanning, which according to our own experience, can provoke aggressive behavior in group-housed male mice due to cross-contamination of odors (etc. urine) between cages. Second, we had previously observed that pharmacological activation of MC1-R reduced plasma cholesterol level and induced hepatic *Ldlr* expression in female *Apoe^-/-^* mice (as discussed on manuscript page 12, lines 475-479), which further supported the preferential use of female mice in the present study.

2) Although immunohistochemistry images in Figure 1A demonstrating the presence of MC1-R in the liver are convincing and understanding that since hepatocytes represent the more abundant cell type in this organ is plausible to conclude that MC1-R is mainly expressed in these cells, these images do not fully support this conclusion. As mentioned in the introduction (lines 84-87), MC1-R has been also demonstrated to be expressed in a variety of peripheral cells. Therefore, to unequivocally demonstrate that MC1-R is mainly expressed in these cells colocalization studies by confocal microscopy of MC1-R and different cell type markers (hepatocytes, macrophages, endothelial cells, fibroblasts…) are required. Without a robust expression of MC1-R exclusively in hepatocytes, further mechanistic studies are not fully justified.

Considering that MC1-R is widely expressed in various tissues and cell types, it is very likely that MC1-R is also present in other cell types of the liver besides hepatocytes. Our initial objective was to assess whether MC1-R is expressed in the liver/hepatocytes (and not to specifically investigate its expression in nonhepatocytes). The presence of MC1-R in the liver/hepatocytes gave us the rationale to engineer the hepatocyte-specific MC1-R KO mouse model to investigate the functional significance of MC1-R in the liver. As suggested by the reviewer, we performed double immunofluorescence staining with the MC1-R antibody in combination with antibodies against cell type-specific markers to further investigate the localization of MC1-R in the liver and to validate its presence in hepatocytes. Importantly, these colocalization studies demonstrate the presence of MC1-R in albumin-expressing hepatocytes (Figure 1B). In addition, MC1-R expression colocalized in cholangiocytes and monocytes/macrophages in the mouse liver (Figure 1—figure supplement 1), but not substantially in endothelial cells (Figure 1—figure supplement 1). We do not however understand the reviewer’s comment: ‘Without a robust expression of MC1-R exclusively in hepatocytes, further mechanistic studies are not fully justified’. Why would MC1-R expression in other cell types nullify the justification for studying the role of MC1-R in hepatocytes? As stated in the introduction (page 3, lines 101-108), our hypothesis was based on the previous findings of disturbed cholesterol and bile acid homeostasis in global MC1-R deficient mice. Therefore, we turned our attention to the liver and more specifically to hepatocytes, which are major regulators of cholesterol homeostasis and bile acid synthesis.

3) Concerns regarding the specificity of the antibody used for both immunohistochemistry images in Figure 1A and western blotting analysis in Figure 1C and Supplementary Figure 1 might rise. As a receptor, MC1-R expected localization would be in the plasma membrane whereas on the presented histological images appears to have uniform immunostaining all over the cytoplasm. Likewise, blots in Supplementary Figure 1 show diverse unspecific bands. In this figure, is the lack of bands in HepG2 protein extract due to the species-specificity of the MC1-R antibody? What is the reason behind the use of mouse heart samples in this study?

We thank the reviewer for these remarks. Regarding Figure 1C and Supplementary Figure 1 (Figure 1—figure supplement 2 in the revised submission), the multiple bands on the Western blot most likely represent native, oligomerized and glycosylated forms of MC1-R as have been previously reported and characterized (*Sánchez-Laorden BL et al., J Invest Dermatol. 2006 Jan;126(1):172-81*). Furthermore, all the major bands disappeared in the presence of molar excess of a blocking MC1-R peptide (Figure 1—figure supplement 2), which indicates that the additional bands were not unspecific. The lack of bands in HepG2 cell extracts in this context might be caused by a considerably difference in the expression level of MC-1R between the samples, since the antibody has been tested to react also with human MC1-R. Mouse heart samples were included as a positive control in Supplementary Figure 3, since they are a rich source of endothelial cells, which are known to express MC1-R. Mouse heart samples have been also used in the antibody testing by other companies (*e.g.* https://www.novusbio.com/products/melanocortin-1-r-mc1rantibody_nbp3-02970).

Regarding the MC1-R immunohistochemistry, it is indeed somewhat unexpected that the positive signal mainly localizes to cytoplasm. However, the appearance of MC1-R positive cells is highly similar with the images provided by the manufacturer (https://www.elabscience.com/p-mc1r_polyclonal_antibody-35168.html) as well as with the images published in other papers (e.g. *Salazar-Onfray F et al., Br J Cancer. 2002 Aug 12;87(4):414-22* and *Cai W at al, Mol Neurodegener. 2022 Feb 23;17(1):16*). It is known that a significant proportion of total MC1-R is internalized and localize to the cytoplasm, particularly in the presence of endogenous/exogenous agonist, which in part explains the expression pattern of MC1-R in the immunohistochemical images.

4) If experimentally possible and/or publicly available data, expression levels of MC1-R in human liver samples from healthy vs. liver disease (hypercholesterolemia, NASH, NAFLD…) will further strengthen your findings and add valuable translational value to your work.

We thank the reviewer for this suggestion, which would indeed add valuable translational value to the study. To this end, we used RNA sequencing data deposited in the Gene Expression Omnibus (GSE120064) to study *MC1R* expression in human liver biopsy samples from patients diagnosed with NAFLD/NASH and from healthy obese control cases (*Govaere O et al., Sci Transl Med. 2020 Dec 2;12(572)*). In good agreement with the findings in Western diet-fed mice, hepatic *MC1R* expression was significantly downregulated.in NAFLD and NASH patients (log2 fold change=-1.1 for both comparisons) (Figure 1E).

5) The observation of that liver from L-MC1-R-/- mice exhibit signs indicative of fibrosis, but no changes were observed in pro-inflammatory genes is intriguing and deserves additional investigation. Does the expression of the fibrosis- and inflammation-related genes tested in the mice changes in the HepG2 under the different treatments employed?

The reviewer raises an important point, which we addressed by analyzing fibrotic and inflammatory gene expression in a-MSH- and LD211-treated HepG2 cells. We found that low concentration (0.1 nM) of LD211 downregulated the expression of fibrotic genes such as *TGFB1*, *ACTA2* and *COL1A1* without affecting the expression of pro-inflammatory genes (Figure 6—figure supplement 1). The concentration-response profile thereby mirrors the changes observed in ERK1/2 and *JNK* phosphorylation (Figure 7B and C, Figure 7—figure supplement 2), which are known to contribute to liver fibrosis (*Jeng K-S et al., Int J Mol Sci. 2020 May 27;21(11):3796* and *Kluwe J et al., Gastroenterology. 2010 Jan;138(1):347-59*). No significant changes were observed in the expression of pro-inflammatory genes. Likewise, low concentration of a-MSH appeared to downregulate the expression of the fibrotic genes *TGFB1* and *COL1A1* (Figure 6—figure supplement 1), but these changes did not reach statistical significance (P=0.08 and 0.051, respectively). These findings suggest that hepatocyte-specific MC1-R deficiency could directly promote liver fibrosis in *L-Mc1r^-/-^* mice as discussed on manuscript page 11/lines 413-417.

6) Liquid chromatography-mass spectrometry data from Figure 3 and Supplementary Figures 4 and 5A should be provided as a supplementary table and raw data deposited in an appropriate public data repository with the corresponding mention in the method section. Moreover, a more detailed description of this methodology would be highly appreciated by the readers.

We thank the reviewer for this suggestion and have now provided the liquid chromatography-mass spectrometry data as source data-files. Furthermore, we added a more detailed description of the BA analysis in the ‘Materials and methods’ section (page 16-17, lines 662-723).

7) What is the biological relevance of the ratio of CA:CDCA observed in Figure 3G and Supplementary Figure 5B?

The ratio of CA:CDCA reflects the contribution of the classical and alternative pathways to bile acid synthesis and is largely determined by the enzymatic activity of CYP8B1. Thus, reduced CA:CDCA ratio in L-Mc1r^-/-^ mice might indicate compromised CYP8B1 activity as discussed on manuscript page 11/lines 424-437.

8) The dysregulation of genes involved in bile acid synthesis, uptake/secretion of bile acids, and nuclear receptors in the liver from KO mice is convincing at the mRNA level in Figures 4A-C. At the protein level (Figures 4D and E) it is unclear. Only two of these proteins were confirmed by western blotting studies (and another protein showed contradictory results). These studies are clearly insufficient in order to draw any valid conclusion. The potential changes of a meaningful number of genes should be validated by alternative methods.

We thank the reviewer for this suggestion. We aimed to validate those genes at the protein level that could potentially explain the observed BA profile in LMc1r^-/-^ mice. It is not uncommon that mRNA level and corresponding protein product of a gene are showing opposite effects. For example, reduced protein level (due to enhanced degradation) could lead to a compensatory increase in mRNA level and the other way around. Therefore, we do not consider the results in Figure 4A and B vs 4D and E to be contradictory, but they rather help to resolve the possible mechanisms leading to disturbed bile acid metabolism in in L-Mc1r^-/-^ mice. We acknowledge that the data does not allow making a conclusion on the exact mechanism, but based on the observed reduction in CA:CDCA ratio and changes in CYP8B1 expression, we speculate that MC1-R deficiency primarily disturbs BA synthesis *via* the classical pathways, leading to compensatory changes in the expression of BA transporters and enzymes involved in the alternative BA synthesis pathway. The results are discussed in more detail on manuscript pages 11/lines 428-445.

9) Blot images presented in Figures 4D, 5A-C, and 5J, and Supplementary Figures 6C-D and 7E do not seem to have quality enough and/or be representative of their corresponding quantifications. Please replace them in accordance.

We appreciate the reviewer’s comment and have now replaced the representative blot images as requested. However, based on the low quality of the blot images obtained with phospho-Akt and total Akt antibodies, we decided to exclude the Supplementary Figures 6D-E and 7E (Figure 7—figure supplement 1 and Figure 7figure supplement 2 in the revised manuscript). This does not affect the conclusions of the manuscript because the underlying mechanisms appear to be independent of Akt phosphorylation.

10) The use of HepG2 cells for the in vitro studies is not enough justified. This cellular model is not ideal for the study of metabolic disturbances and does not match some of the criteria used in the in vivo model (they are derived from a male child with hepatocellular carcinoma). It may be understood that they were used for initial screening purposes but selected, key aspects, of the study should be checked in primary hepatocytes from L-MC1-R-/- and control mice.

We agree with the reviewer that HepG2 cells are not an ideal in vitro model to study certain aspects of hepatocyte biology. Despite the apparent differences between HepG2 cells and primary hepatocytes, comparative studies have demonstrated that these models have similar properties in terms of cholesterol metabolism (*Nagarajan SR et al., Am J Physiol Endocrinol Metab. 2019 Apr 1;316(4):E578-E589* and *Wang SR et al., Biochim Biophys Acta. 1988 Aug 12;961(3):351-63*). More importantly, the central model of the study is the hepatocyte-specific MC1-R deficient mice that were used to test the hypothesis that MC1-R is expressed in hepatocytes and involved in the regulation of cholesterol metabolism. Having noted a phenotype of enhanced lipid accumulation in L-Mc1r^-/-^ mice, we proceeded to investigate the direct actions and underlying mechanisms of aMSH and the selective MC1-R agonist in HepG2 cells. Without the in vivo evidence, it would be more critical to investigate whether the key responses observed in HepG2 cells are also recapitulated in primary hepatocytes.

We originally planned to address the reviewer’s comment by isolating primary hepatocytes from *L-Mc1r^-/-^* and control mice. However, after producing new mice for the in vivo experiment (Figure 2—figure supplement 2), we have had major difficulties in expanding and even maintaining this mouse colony. Therefore, we have not been able to perform these experiments within a reasonable time frame.

11) The decrease in MC1-R in HepG2 in response to palmitic acid, but not when exposed to LDL or atorvastatin, is an interesting observation. Based on what did you choose the selected concentration of these compounds to mimic the cholesterol-rich diet administrated to the animals?

The concentrations of palmitic acid, LDL and atorvastatin were selected based on previous publications that have demonstrated these concentrations to evoke significant biological responses in terms of lipid and cholesterol loading (*Hae Seo M et al., PLoS One. 2016 Dec 1;11(12)* and *Feng J et al., Commun Biol. 2019 May 8;2:173.*) and cholesterol depletion (*e.g. Mohammadi A et al., Arterioscler Thromb Vasc Biol. 1998 May;18(5):783-93*.), respectively. The purpose was not to specifically mimic the cholesterol-rich diet administered to the animals, because it is highly challenging to estimate the exposure of hepatocytes in vivo to different components of the diet. Furthermore, even if the exposure could be estimated, HepG2 cells might respond to the treatments at a different concentration compared to primary hepatocytes.

12) The authors also show that α-MSH treatment reduced the lipid metabolism in hepatocytes. However, it is unclear whether the pathological mechanisms mediated by the downregulation of MC1-R in hepatocytes would be restored by the addition of α-MSH. Additional studies in which MC1-R expression can be manipulated may be needed. Similarly, it would also be worthwhile to use clinically approved melanocortin receptor-targeted drugs in these experiments.

We thank the reviewer for these suggestions. We do not however understand what the reviewer means by ‘*it is unclear whether the pathological mechanisms mediated by the downregulation of MC1-R in hepatocytes would be restored by the addition of* a*-MSH*.’ MC1-R deficiency/downregulation in hepatocytes is likely to disturb the signaling mediated by the endogenous MC1-R agonist a-MSH as well as to lead to pathological changes even in the absence of aMSH, considering that MC1-R possesses agonist-independent constitutive activity (*e.g. Sánchez-Más J et al., Pigment Cell Res. 2004 Aug;17(4):386-95*). The addition of a-MSH to a MC1-R deficient model is likely to evoke signaling *via* other MC-R subtypes (and MC1-R in other tissues/cell types) and could thus potentially restore some pathophysiological mechanisms caused by hepatocyte-specific MC1-R deficiency/MC1-R downregulation. We do not however anticipate that addition of aMSH would restore MC1-R expression/activity in hepatocytes and therefore, this kind of experiment would not help in unraveling the role of MC1-R signaling in hepatocytes. Likewise, the use of clinically approved melanocortin receptor-targeted drugs in these experiments would involve significant uncertainty in evaluating the contribution of MC1-R-dependent signaling on the observed effects, since none of these drugs is selective for MC1-R.

13) The attempt to elucidate the intracellular signalling cascades that might be behind the beneficial effects of MC1-R in the liver is mostly appreciated. Nevertheless, in the present form, these investigations fail to pinpoint a specific mechanism leading to inconclusive results. It would be more convincing if the authors can perform transcriptomics or proteomics in primary hepatocytes from L-MC1-R-/- and control animals to pursue potential targets.

As suggested by the reviewer, we performed transcriptomics/RNA sequencing analysis for liver samples from L-Mc1r^-/-^ and control mice. Unfortunately, we did not have sufficient number of mice with desired genotypes to be used for primary hepatocyte isolation and subsequent transcriptomics. Thus, we needed to perform transcriptomics using previously collected liver samples. A Volcano plot of differentially expressed genes (DEGs) in the liver of chow-fed L-Mc1r^-/-^ (n=4) versus L-Mc1r^+/-^ mice (n=4) is shown in Author response image 2. The x-axis shows log2 fold changes (logFC) between genotypes, the y-axis plots negative log10 P-values.

Among the most significant DEGs, apolipoprotein A-IV (*Apoa4*) and sortilin-related receptor, LDLR class A (*Sorl1*) were upregulated (FDR-corrected P<0.0001) in the liver of chow-fed L-Mc1r^-/-^ mice. These DEGs were also validated by qPCR (Author response image 3).

**Author response image 3. sa2fig3:** 

*Apoa4* gene is known to be expressed in the intestine and implicated in lipid absorption. In rodents, *Apoa4* is also expressed in the liver and robustly upregulated after experimentally induced hepatic steatosis (*VerHague MA et al., Arterioscler Thromb Vasc Biol. 2013 Nov;33(11):2501-8*). Therefore, the upregulation of *Apoa4* in *L-Mc1r^-/-^* mice might be a consequence rather than driver of increased lipid accumulation in the liver. *Sorl1*, on the other hand, has been implicated in the pathogenesis of Alzheimer’s disease, but also in metabolic regulation. Overexpression of *Sorl1* in adipocytes inhibits release of free fatty acids and enhances fat deposition (*Schmidt V et al., J Clin Invest. 2016 Jul 1;126(7):2706-20*). However, the role of *Sorl1* in the liver is unknown. In addition, gene set enrichment analysis (GSEA) for gene ontology revealed only one significant term: extracellular space (adjusted Pvalue: 0.0071, number of genes: 55).Taken together, the results of RNA-seq analysis do not provide clear answers that would help in identifying potential targets that could explain the observed phenotype in L-Mc1r^-/-^ mice. Therefore, we would rather omit these data from the main manuscript.